# Impacts of tropical cyclones on the thermodynamic conditions in the tropical tropopause layer observed by A-train satellites

Jing Feng[1] and Yi Huang[1]

[1]Department of Atmospheric and Oceanic Sciences, McGill University, Montreal, Canada

**Correspondence:** Jing Feng (jing.feng3@mail.mcgill.ca)

**Abstract.** The tropical tropopause layer (TTL) is the transition layer between the troposphere and the stratosphere. Tropical cyclones may impact the TTL by perturbing the vertical distributions of cloud, temperature, and water vapor. This study combines several A-Train instruments, including radar from CloudSat, lidar from Cloud-Aerosol Lidar and Infrared Pathfinder Satellite Observations (CALIPSO) satellite, and the Atmospheric InfraRed Sounder (AIRS) on Aqua satellite, to detect signatures of cyclone impacts on the distribution patterns of cloud, water vapor, temperature, and radiation by compositing these thermodynamic fields relative to cyclone center location.

Based on the CloudSat 2B-CLDCLASS-LIDAR product, this study finds that tropical cyclone events considerably increase the occurrence frequencies of TTL clouds, in the form of cirrus clouds above a clear troposphere. The amount of TTL cloud ice, however, is found to be mostly contributed by overshooting deep convection that penetrates the base of the TTL at 16 km.

To overcome the lack of temperature and water vapor products in cloudy conditions, this study implements a synergistic method that retrieves temperature, water vapor, ice water content, and effective radius simultaneously by incorporating observations from AIRS, CloudSat, and CALIPSO. Using the synergistic method, we find a vertically-oscillating pattern of temperature anomalies above tropical cyclones, with warming beneath the cloud top (around 16 km) and cooling above. Based on water vapor profiles retrieved by the synergistic method, we find that the layer integrated water vapor above 16 km (LIWV) is higher above tropical cyclones, especially above overshooting deep convective clouds, compared to climatological values.

Moreover, we find that the longwave and net radiative cooling effect of clouds prevails within 1000 km of tropical cyclone centers. The radiative heating effects of clouds from the CloudSat 2B-FLXHR-LIDAR product are well differentiated by the collocated brightness temperature of an infrared window channel from the collocated AIRS L1B product. By performing instantaneous radiative heating rate calculations, we further find that TTL hydration is usually associated with radiative cooling of the TTL, which inhibits the diabatic ascent of moist air across isentropic surfaces to the stratosphere. Therefore, the radiative balance of the TTL under the impact of the cyclone does not favor the maintenance of moist anomalies in the TTL or transporting water vertically to the stratosphere.

## 1 Introduction

The tropical tropopause layer (TTL) is the transition layer between the convective overturning circulation in the troposphere and the Brewer-Dobson circulation in the stratosphere. Once entering the TTL, air tends to rise into the stratosphere, driven

by extratropical wave-drag and influenced by local radiative heating (Holton et al., 1995; Corti et al., 2006). The clear-sky radiative heating becomes positive above the level of zero radiative heating (LZRH). This level marks the altitude where the ascending motion becomes prevailing.

The TTL plays an important role in stratosphere-troposphere exchange (Holton et al., 1995). For example, the low temperature in the TTL act as a 'cold trap' that modulates both the vertical and isentropic (quasi-horizontal) transport of water vapor to the lower stratosphere (Dessler et al., 1995; Brewer, 1949; Holton and Gettelman, 2001; Gettelman et al., 2002), where water vapor, despite its low concentration, may have a large impact on radiation, climate, and atmospheric chemistry (Solomon et al., 2010; Anderson et al., 2012; Dessler et al., 2013; Huang et al., 2016).

Given its vertical extent, deep convection potentially provides an important pathway to transport water vapor and other constituents to the stratosphere via the TTL. Deep convection may affect the TTL in several ways. First, tropical deep convection, especially in tropical cyclones, is associated with strong dynamical cooling around the tropopause level (Holloway and Neelin, 2007). Second, the injected ice and water vapor, together with the temperature anomalies caused by deep convection, can modify the radiative heating in the TTL, which in turn can either speed up or slow down the upwelling motion of air and the transport to the stratosphere. Consequently, air convectively injected into the TTL can be either a source or a sink of water vapor, depending on the pre-existing relative humidity (Ueyama et al., 2018; Schoeberl et al., 2018). Simulations and observations have shown that deep convection may hydrate the upper-troposphere and lower-stratosphere (UTLS) by directly injecting water vapor and ice above mid-latitude (Anderson et al., 2012; Sun and Huang, 2015; Qu et al., 2020) and tropical storms (Avery et al., 2017; Schoeberl et al., 2018), or dehydrate it by condensing the pre-existing water vapor to ice particles in supersaturated environments (Ueyama et al., 2018).

On the other hand, climate models and global reanalysis datasets are subject to common problems in representing key processes, such as convective parameterization, (e.g., Takahashi et al., 2016), in the UTLS region. These problems include a persistent wet bias in upper tropospheric humidity (Huang et al., 2007; Jiang et al., 2012, 2015), discrepancies in the transportation speed of water vapor from the upper troposphere to the lower stratosphere (Jiang et al., 2015; Schoeberl et al., 2012) and contradictory assessments of cloud impacts on diabatic heating in the TTL region (Wright and Fueglistaler, 2013; Wright et al., 2020).

Existing satellite datasets (Waters et al., 2006; Bernath et al., 2005; Anthes et al., 2008) and aircraft campaigns (e.g., Jensen et al., 2013; Lee et al., 2019) have advanced understanding of the TTL region, although the study of deep convective impacts on temperature, water vapor, and clouds in the TTL region is still impeded by a lack of collocated measurements of these variables. The A-Train constellation (L'Ecuyer and Jiang, 2011) carries over 20 instruments that monitor clouds and other atmospheric variables. However, the sounding of thermodynamic conditions above deep convection, especially near the convective core, remains a challenge (Livesey et al., 2017; Olsen et al., 2013). Feng and Huang (2018) found that the retrievability of temperature and water vapor is improved by an underlying cloud layer because the cloud layer reduces the degeneracy caused by non-monotonic vertical temperature variations and the smearing effect of lower-level water vapor, and proposed a cloud-assisted retrieval algorithm that can be applied to infrared hyperspectral measurements, such as those from the Atmospheric Infrared Sounder (AIRS, Chahine et al., 2006) aboard Aqua (Parkinson, 2003). Feng et al. (2021) further developed a synergistic method

that incorporates cloud measurements of collocated active cloud profilers. By conducting a simulation experiment, Feng et al. (2021) demonstrated that this method can capture the variability of temperature and humidity above tropical convective storms and improve the retrievals near the cloud top through the incorporation of information from active sensors.

In this study, we aim to quantify the effect of tropical cyclones on TTL temperature, water vapor, and clouds using A-Train satellite observations, specifically hyperspectral infrared measurements from AIRS and cloud profiles from CloudSat (Stephens et al., 2008) and CALIPSO (Cloud, Aerosol Lidar and the Infrared Pathfinder Satellite Observations; Winker et al., 2010). Tropical cyclones are of particular interest here because they constitute a large fraction of the most energetic (overshooting) convective clouds in the tropics (Romps and Kuang, 2009) and provide vertically extended dense high clouds that enable the above-cloud temperature and humidity retrieval method developed by Feng and Huang (2018) and Feng et al. (2021). By using satellite observational datasets, which are introduced in Section 2, we aim to understand: 1) how tropical cyclones, especially the overshooting events, impact TTL cloud occurrence and cloud ice (see Section 3.1 and 3.2), 2) whether tropical cyclones lead to an overall hydration in the TTL (see Section 3.3), and 3) how tropical cyclones affect radiative heating in the TTL (Section 4). These questions are further discussed together with key conclusions in Section 5.

## 2   Data and Methodology

### 2.1   Datasets

Following a sun-synchronized orbit with a repeat-cycle of 16 days, the A-Train satellites cross the equator at around 1:30 pm solar time in the ascending nodes and 1:30 am in the descending nodes every day.

CloudSat uses a cloud profiling radar (CPR) that operates at 94-GHz to observe cloud and precipitation. Sampling along-track at every 1.1 km, this instrument has a cross-track resolution of 1.4 km and along-track resolution of 1.8 km. The Cloud-Aerosol Lidar with Orthogonal Polarization (CALIOP) is a polarization lidar carried by CALIPSO that operates at 532 and 1064 nm. The sensitivities of the two instruments in detecting cloud properties differ. The lidar signals are sensitive to cloud particles at all sizes but are quickly attenuated in thick clouds. The radar is not sensitive to clouds with effective particle sizes smaller than 40-50 microns, but can observe thick clouds and precipitation in storm cores. Hence, it is essential to combine the lidar and radar to obtain the sensitivity to both optically thin cirrus in the TTL and optically thick deep convective clouds (DCC).

CloudSat provides several radar-lidar products by merging coincident lidar pulses within the CPR footprints (Mace et al., 2009). In this study, we use the cloud classification (2B-CLDCLASS-LIDAR, Sassen et al., 2009) and radiative heating rates (2B-FLXHR-LIDAR, Henderson et al., 2013) products (version P2_R04). In the 2B-CLDCLASS-LIDAR product, eight cloud types are classified, including cirrus, altostratus, altocumulus, stratus, stratocumulus, cumulus, nimbostratus, and deep convective clouds, depending on the vertical distribution of hydrometeors inferred from radar signal intensity and also their horizontal length scales (Wang et al., 2012). DCCs are classified based on several conditions, including a vast horizontal and vertical extent, dense hydrometers (as inferred from radar reflectivity near the cloud top), and also the presence of precipitation (Wang et al., 2012). The heating rate profiles in 2B-FLXHR-LIDAR are derived from two-stream broadband radiative transfer cal-

culations combining the 2B-CWC-RO cloud water content profile, collocated CALIPSO version-3 products (Trepte et al., 2010), and atmospheric state profiles (temperature, water vapor, and ozone) from ECMWF forecasts, which are included in the ECMWF-AUX product (Partain, 2004).

Operational CloudSat-CALIPSO synergistic algorithms have been developed to retrieve ice cloud properties (Austin et al., 2009; Deng et al., 2010; Okamoto et al., 2010; Delanoë and Hogan, 2010) and are evaluated (Deng et al., 2013; Saito et al., 2017). This study uses a DARDAR-Cloud (raDAR/liDAR; Delanoë and Hogan, 2010) product v2.1.1 which combines coincident lidar attenuated backscatter measurements (from CALIPSO Level 1 v3) and radar reflectivity (from 1B-CPR R04). Based on an optimal estimation method (Rodgers, 2000), DARDAR iteratively retrieves the state vector which contains visible extinction, lidar extinction-to-backscatter ratio, and particle number concentration and converts it to ice water content (IWC) and effective radii profiles at each CPR footprint with horizontal (cross-track) resolution of 1.4 km and a vertical resolution of 60 m following a set of cloud microphysical parameterizations (i.e., particle size distribution and mass-size relation). This study uses IWC profiles from DARDAR but cloud type classified by the CloudSat 2B-CLDCLASS-LIDAR product. Due to differences in retrieval algorithms and cloud microphysical assumptions (Deng et al., 2013), there are discrepancies between the two products in terms of ice cloud occurrence (McErlich et al., 2021). For consistency, we only use footprints where both radar and lidar observations are available and treat a high cloud layer detected by DARDAR but not classified in 2B-CLDCLASS-LIDAR as the same cloud type as its adjacent cloud layer, or as cirrus if it is isolated.

The Aura Microwave Limb Sounder (MLS, Waters et al., 2006) retrieves temperature and trace gases for pressure levels less than 316 hPa, with a vertical resolution of around 3 km for water vapor and 4.7 km for temperature. The documented accuracy of the version 4.2 product at the level of 100 hPa is 8% for water vapor and 0.7 K for temperature. Although MLS can retrieve atmospheric states in moderately cloudy conditions, line shape distortion caused by the strong scattering of thick clouds limits the retrieval capability (Livesey et al., 2017). Therefore, only data not affected by clouds, based on the *status* flag included with the product, are used to avoid degraded data quality. Moreover, due to the limb-viewing scanning geometry, MLS has a relatively large sampling footprint with a horizontal resolution around 200 km along the track, which limits its sensitivity to small-scale variability (Schwartz et al., 2013).

AIRS measures infrared spectra from 650 $cm^{-1}$ to 2665 $cm^{-1}$ with 2378 channels, using cross-track scans to provide large spatial coverage. Only the fields-of-view (FOVs) with viewing angles within 15°of the nadir is used in this study, considering that the limb-view geometry increases the optical depth and the atmospheric attenuation. The selected viewing angle corresponds to a cross-track span of around 400 km. The high spectral resolution in the mid-infrared makes AIRS sensitive to temperature, water vapor, and also clouds. However, the standard AIRS retrieval is not sensitive to the water vapor signal from the UTLS (Fetzer et al., 2008; Gettelman et al., 2004; Read et al., 2007) because of the weak absorption/emission of the dry UTLS relative to the troposphere and a cloud-clearing algorithm (Susskind et al., 2003) adopted by the AIRS standard retrieval. This algorithm infers temperature and trace gases of the clear-sky portion of adjacent 3×3 FOVs with varying cloud amounts by contrasting the nine FOVs, assuming a uniform distribution of these atmospheric states. Consequently, it degrades the horizontal resolution by a factor of three and suffers from large uncertainties when cloud amounts are uniform or when temperature and trace gases change drastically among the nine FOVs.

Therefore, instead of using the AIRS Level 2 product for above-cloud atmospheric conditions, we apply a synergistic, cloudy-sky retrieval method developed from the cloud-assisted method proposed by Feng and Huang (2018). Targeting at dense high-level clouds, this retrieval method retrieves water vapor, temperature, ice water content, and effective radius from the AIRS L1B product by combining active observations of mass concentration and effective size of ice cloud particles from the DARDAR-Cloud product. This approach is hereafter referred to as either a joint AIRS-DARDAR retrieval or a synergistic retrieval. While the information on temperature and water vapor profiles above clouds is obtained from spectrally-dependent optical depths, the inclusion of active cloud observations substantially reduces uncertainties in cloud top position and hence increases the sensitivity of the retrieval to the temperature and water vapor of topmost cloud layers. Using the wavenumber-dependent cloud extinction coefficients in the mid-infrared channels, this method marginally updates the concentration and effective particle size of ice clouds relative to the collocated cloud observations of active sensors. The effective radius is estimated as a vertically averaged value that produces the most reasonable mid-infrared emission spectra of thick high-level clouds (Feng et al., 2021). Through a simulation experiment, Feng et al. (2021) demonstrated that the synergistic method is sensitive to spatial variability in thermodynamic conditions above deep convection. Details of this retrieval method are presented in Appendix A. The retrieval achieves a precision of 0.31 K and 0.36 ppmv for temperature and water vapor, respectively, as illustrated by Fig. A2. The vertical resolution at 100 hPa is 3.2 km for temperature and 5.8 km for water vapor.

The impact of cyclones is assessed by calculating anomalies in cloud and non-cloud variables compared to their climatological values. In this paper, we define the climatology as the multi-year monthly mean of variables from 2006 to 2016, using IWC from DARDAR v2.1.1, temperature from AIRS L2 v6, water vapor from MLS v4.2, and brightness temperatures at 690 and 1231 $cm^{-1}$ derived from AIRS L1B v5. All variables are assessed on a $1° \times 1°$ longitude-latitude grid unless specified otherwise.

## 2.2 Compositing method

In this study, a list of tropical cyclones observed by the A-Train satellites is obtained from the CloudSat tropical cyclone product (2D-TC, Tourville et al., 2015). This product uses best-track information provided from the Automated Tropical Cyclone Forecasting System (Tourville et al., 2015; Sampson and Schrader, 2000) to identify the cyclone center position. Note that only daytime measurements are available after April 17th, 2011, due to a spacecraft battery issue. In addition to CloudSat, we combine CALIPSO, MLS, and AIRS together to provide the cloud distributions and atmospheric states associated with each cyclone overpass.

Measurements are composited relative to cyclone center locations in the northern part of the West-Pacific region (the boxed region in Fig. 1 (a)), given the abundance of data samples in this region. The density of measurement locations for each instrument is shown in Fig. 1. Samples from AIRS (Fig. 1 (d)) are of higher density compared to CloudSat (Fig. 1 (b)) and MLS (Fig. 1 (c)) owing to the advantage of the cross-track scanning of AIRS. Considering the differences in the sample densities and FOV sizes of the original measurements of these instruments, the cyclone-centered composites are constructed by averaging variables over different spatial scales: 60 km (CloudSat, as well as DARDAR which reports at CloudSat footprints), 120 km (MLS), and 20 km (AIRS), to ensure sufficient samples are obtained. Only observations collected over the ocean are

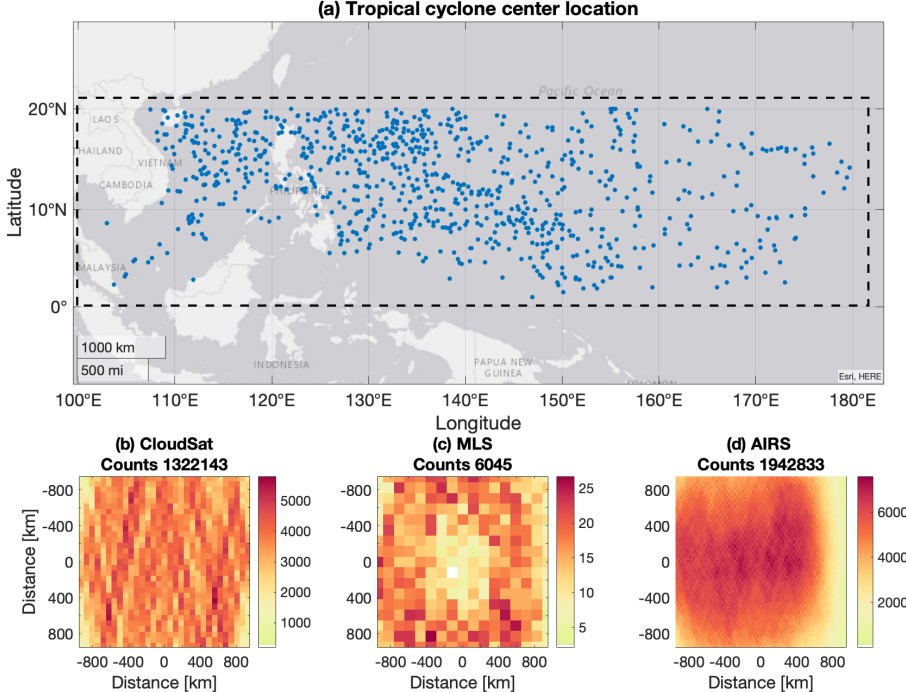

**Figure 1.** Distributions of cyclone centers passed over by A-Train and sample density from A-Train instruments with respect to cyclone center locations. (a) Locations of tropical cyclones centers (947 in total) that passed over by the A-Train satellites from 2006 to 2016 over the northern part of the West-Pacific region (within the boxed area). The sample density of (b) CloudSat (the DARDAR product is available at the horizontal footprints of CloudSat), (c) MLS, and (d) AIRS (limited to viewing angles within 15°of nadir) with respect to distances to cyclone center locations. The sample densities are defined as the number of samples per 100 km×100 km area, measured by normalizing sample counts within every 60 km (b), 120 km (c), and 20 km (d) grid box over the area of the grid box. The numbers on the top of each panel show the total number of samples.

used to avoid sample discrepancies arising from land-sea contrast. While both daytime and nighttime measurements are used, fewer nighttime measurements are available because of the daylight-only operation of CloudSat after 2011. Diurnal differences

are noticed in IWC profiles and the brightness temperature in the 2006-2011 period but are not addressed in study.

## 3    Tropical cyclone impacts

### 3.1    Cloud distribution

The datasets introduced earlier are used to depict the cloud distributions above cyclones. DCCs and TTL clouds are especially of interest here. TTL clouds, for the convenience of the analysis, are defined as cloud columns with one or more cloud layers

above 16 km, where the clear-sky LZRH and the mean tropopause (WMO, 1957) height typically locates in the tropics (Yang

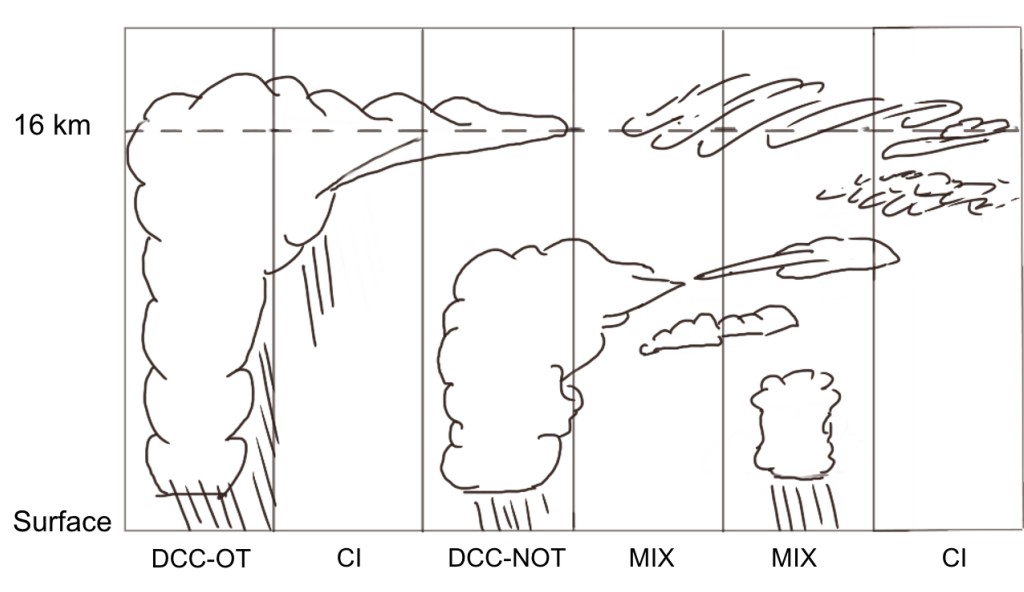

**Figure 2.** A schematic of the TTL cloud categories (see definitions in Section 3.1).

et al., 2010). These TTL clouds are further broken into four categories to distinguish TTL clouds with or without underlying deep convective clouds. As shown by the schematic in Fig. 2, the four cloud categories are defined as follows:

- DCC-OT: deep convective clouds whose top boundary exceeds 16 km.

- DCC-NOT: deep convective clouds whose top boundary is below 16 km while non-convective clouds are detected above 16 km.

- CI: cirrus detected above 16 km and no DCC or any other cloud in the column.

- MIX: the remaining conditions (no DCC, but cloud detected above 16 km).

The DCC-OT category is used to identify continuous convective clouds that extend above 16 km, which is the average altitude of the 380 K isentropic surfaces in the region of interest. These clouds are referred to as "overshooting" DCCs in the context of this paper. For convenience, TTL cloud categories without overshooting, including DCC-NOT, CI, and MIX, are also grouped as TTL-OTHER. Clouds with their top boundary below 16 km are referred to as NTTL.

As depicted by Fig. 2, cloud columns with DCCs are classified as either DCC-OT or DCC-NOT, where the former refers to overshooting DCCs and the other refers to any other cloud types that contain DCCs in the column. The two categories are

distinguished by whether the lowermost TTL clouds are connected with underlying DCCs because the adjacent clouds are essentially considered as the same cloud type in the CloudSat-2B-CLDCLASS-LIDAR product. DCC-NOT typically consists DCCs and cirrus with possible middle cloud layers between DCCs and cirrus. In the MIX category, TTL clouds may lie over either middle clouds (accounting for $90\%$ of the MIX category) or low clouds ($10\%$ of the MIX category). Columns with only cirrus clouds are classified as CI. MIX and CI categories also include optically thick anvil clouds near the edge of DCCs, because anvil cloud is classified as cirrus or altostratus in the 2B-CLDCLASS-LIDAR (Wang et al., 2012; Young et al., 2013), depending on its vertical position.

The occurrence frequency of clouds and four cloud categories is then calculated as the ratio between the number of samples with a certain feature, e.g., TTL clouds, and the number of overpass samples in each $60 \text{ km} \times 60 \text{ km}$ grid box in the cyclone-centered composite domain (Fig. 3 (a-b,d-g)). Using IWC profiles from DARDAR-Cloud, a composite of ice water path (IWP) is derived in the same grid boxes. These results are also shown as a function of radial distance from the cyclone center in Fig. 4 (a-c).

Figures 3 (a) and Fig. 4 (a) show that TTL clouds occur frequently above tropical cyclones. In the $2000 \text{ km} \times 2000 \text{ km}$ cyclone-centered composite domain, TTL clouds have an occurrence frequency of 0.37 on average, which is significantly greater than the climatological value of 0.03. This climatological value is derived from the DARDAR-Cloud from 2006 to 2016, regardless of the presence of cyclones. DCCs occur mostly within $400 \text{ km}$ of the cyclone center (Fig. 4 (a)) and are noticeably more often on the southwest side of tropical cyclones (Fig. 3 (b)). The occurrence frequency of DCCs in the composite domain is 0.1 on average, while the climatological value (regardless of cyclone condition) is only 0.008. These results suggest that tropical cyclones considerably increase the occurrence frequencies of both DCCs and TTL clouds.

Figures 3 (d-g) and 4 (b) break down the occurrence frequencies of different TTL cloud categories. They show that DCC-OT tends to occur in the southwest quadrant within the $400 \text{ km}$ radius. Outside the quadrant, the CI category has an occurrence frequency generally over 0.2 which makes up over $60\%$ of the total TTL cloud occurrence. It suggests that the formation of high clouds are greatly promoted by tropical cyclone events, possibly by 1) transport of cloud ice via convective outflow from cyclone centers and propagating waves and 2) by local cooling that condensates supersaturated water vapor into ice (Tseng and Fu, 2017; Schoeberl et al., 2019) as a result of dynamical and radiative cooling which is discussed in Sections 3.3.1 and 4.

In contrast to the uniformly distributed TTL cloud (Fig. 3 (a)), Fig. 3 (c) shows that the TTL cloud ice is concentrated in regions closest (within $200 \text{ km}$) to the cyclone center, most often to the southwest, similar to the DCC-OTs (Fig. 3 (d)). This coincidence suggests potential contributions from DCCs that penetrate $16 \text{ km}$ (DCC-OTs) to TTL cloud ice. Hence, we calculate the fractional contribution to TTL cloud ice from each TTL cloud category, defined as the ratio between the sum of TTL cloud ice of one cloud category and the total TTL cloud ice. Figure 4 (c) confirms that DCC-OTs account for over $80\%$ of TTL cloud ice near the cyclone center. After integrating over the area within $1000 \text{ km}$ of cyclone centers, DCC-OT accounts for $43\%$ of the cloud ice with only $5\%$ of the observed TTL cloud over, while CI accounts for $23\%$ of the cloud ice with $48\%$ of the observed TTL cloud cover. Overall, Fig. 3 and 4 suggest that most TTL clouds above cyclones are cirrus above a clear troposphere (CI), while the TTL cloud ice is largely contained by overshooting deep convective clouds (DCC-OTs).

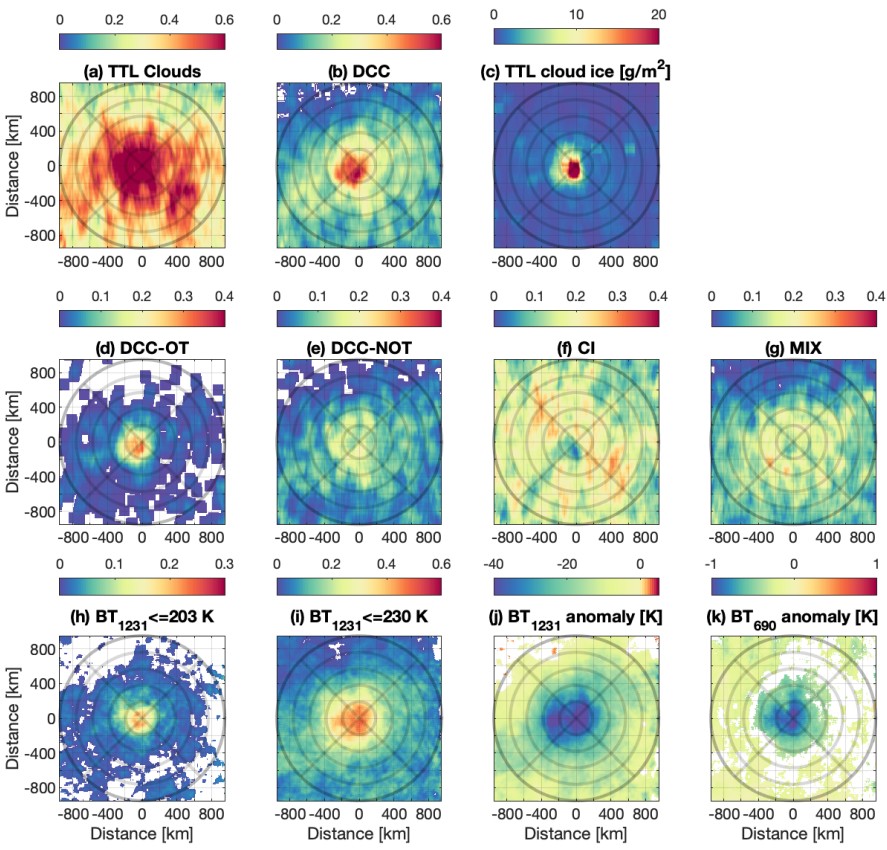

**Figure 3.** Cyclone-centered composite of cloud statistics. (a) The occurrence frequency of TTL clouds (i.e., clouds above 16 km) and (b) deep convective clouds (regardless of TTL cloud occurrence). (c) Ice water path ($g/m^2$) above 16 km. The occurrence frequency of four TTL cloud categories (defined in Section 3.1 and schematically shown in Fig. 2): (d) DCC-OT (overshooting DCCs), (e) DCC-NOT (non-overshooting DCCs), (f) CI (cirrus), and (g) MIX (remaining conditions). The occurrence frequency of clouds identified by infrared radiance measurements: (h) deep convective clouds with $BT_{1231} <= 203$ K and (i) overcast high clouds with $BT_{1231} <= 230$ K. (j) Brightness temperature anomalies [K] in an atmospheric window channel ($BT_{1231}$) and (k) a $CO_2$ channel ($BT_{690}$). Upper (a-c), middle(d-g), and lower (h-k) panels are based on data from DARDAR-Cloud-IWC, CloudSat-2B-CLDCLASS-LIDAR, and the AIRS L1B v5 product, respectively. Only statistically significant occurrence frequencies (at a 99% confidence level, compared to zero) are shown in (a,b,d-i), and only significant brightness temperature anomalies (99%, compared to the climatology) are shown in (j,k).

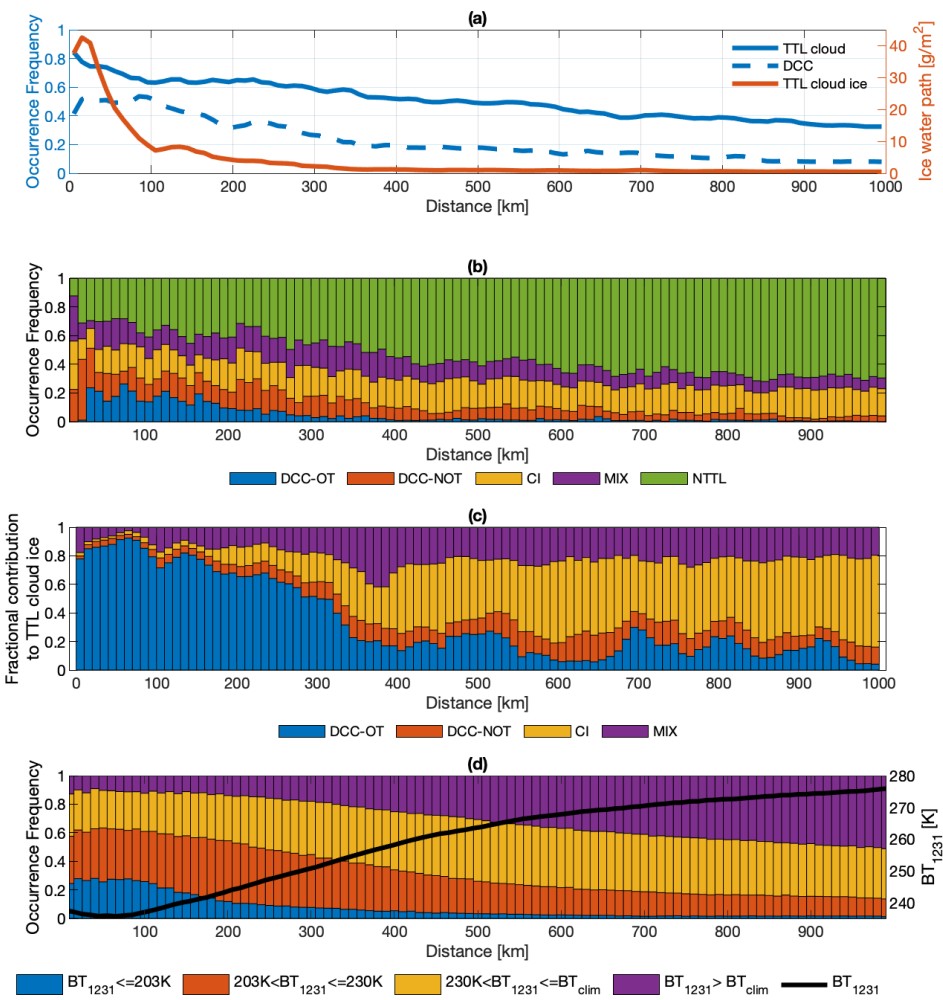

**Figure 4.** Cloud statistics as a function of radial distance to cyclone center. (a) Occurrence frequency of TTL clouds (blue solid curve) and deep convective clouds (blue dashed curve); the red curve is the ice water path $(g/m^2)$ above $16\,\mathrm{km}$. (b) Occurrence frequency of each cloud category (schematically shown in Fig. 2). (c) Fractional contribution to the TTL cloud ice (the red curve in panel (a)) by each cloud category. (d) Occurrence frequencies of clouds classified by $BT_{1231}$. $BT_{clim}$ refers to the multi-year monthly mean $BT_{1231}$. The black curve shows the average $BT_{1231}$.

## 3.2 Infrared Radiance

In the previous section, the distribution patterns of clouds above tropical cyclones are analyzed by combining observational products from CloudSat and CALIPSO. Compared with the nadir-view-only instruments, AIRS provides expanded spatial coverage by performing a cross-track scan. With over two thousand channels, this hyperspectral instrument can profile atmospheric absorbers and temperatures. For example, the brightness temperature (BT) in the atmospheric window channel is a measure of cloud thermal emission temperature which is useful for inferring cloud top height.

As described in Section 3.1, the overshooting DCC is a major source of TTL cloud ice. The spectral signatures of the overshooting DCCs are further investigated. Previous studies (Aumann et al., 2011; Aumann and Ruzmaikin, 2013) have shown that overshooting DCCs are identifiable by cold BT anomalies in the window channels, e.g., at the 1231 $cm^{-1}$ channel ($BT_{1231}$) and also positive BT difference between the water vapor and window channels ($\Delta BT = BT_{1419} - BT_{1231}$), (Aumann et al., 2011; Aumann and Ruzmaikin, 2013). The cold $BT_{1231}$ is a result of the extremely high vertical reach (and thus the very low cloud top temperature) of the DCCs; the positive $\Delta BT$ indicates emission from warm stratospheric layers against the cold cloud top. However, there is no consensus on the threshold values of these quantities to identify DCCs, partly because of uncertainty in the temperature distribution above DCCs due to the impacts of convection. Following the statistical analysis detailed in Appendix B, we find the optimal threshold for identifying overshooting DCCs to be $BT_{1231} <= 203$ K, corresponding to a false positive rate of 0.008 and a false negative rate of 0.323. We find that incorporating $\Delta BT$ does not improve the detection.

This $BT_{1231} <= 203$ K criterion is then applied to AIRS overpass measurements (Fig. 1 (d)). Figure 3 (h) shows that the identified deep convective clouds by this criterion are distributed similarly to the overshooting DCC (DCC-OT) classified by CloudSat. It also confirms that overshooting DCCs prefer to occur in the southwest quadrant.

Similarly, a $BT_{1231} <= 230K$ criterion is used to identify thick upper-tropospheric clouds. This criterion detects cloud tops above 11 km, which corresponds to a climatological mean temperature of 230 K. The identified upper-tropospheric clouds are distributed mainly within the 400 km radius, as depicted in Fig. 3 (i), with a frequency of more than 0.3. Figure 3 (i) also reveals fewer thick high clouds on the northwestern quadrant of the domain, similar to the results based on CloudSat data (Fig. 3 (e,g)). This $BT_{1231}$ criterion is also used for identifying FOVs with thick upper-tropospheric clouds to perform the synergistic retrieval method discussed in Section 3.3.

For each AIRS overpass sample, the $BT_{1231}$ anomaly is calculated as the deviation from the climatology. As shown in Fig. 3 (j), cyclones induce a significant cold anomaly in $BT_{1231}$ over the composite domain.

The reduced window channel radiance ($BT_{1231}$) suggests that cyclone clouds effectively attenuate infrared radiation emitted from the surface, thus potentially leading to a net radiative cooling of the atmospheric layers above the clouds. Hence, it is interesting to examine whether tropical cyclones leave detectable signatures in the temperature fields. A composite of BT anomalies using a $CO_2$ absorption channel (690 $cm^{-1}$) is shown in Fig. 3 (k). This channel has a weighting function that peaks at 85 hPa and is thus sensitive to the cold point temperatures (i.e., the vertical temperature minima) that climatologically occurs near this level. Indeed, Fig. 3 (k) shows a cold $BT_{690}$ anomaly above cyclones, especially around the cyclone center.

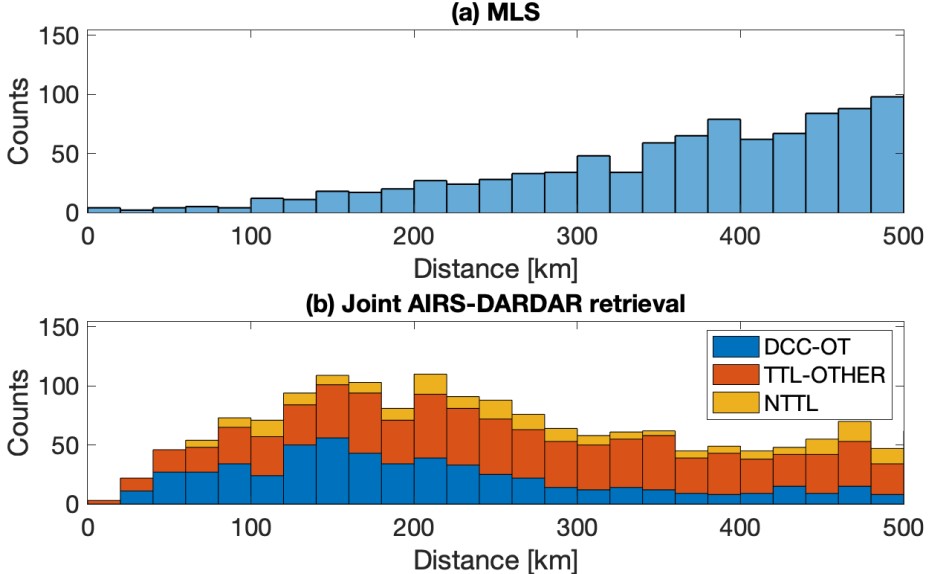

**Figure 5.** Sample densities used for assessing temperature and humidity distributions from (a) MLS and (b) the joint AIRS-DARDAR retrieval.

However, we cannot eliminate the possible impacts of cloud emission, which will be addressed in the following section using different methods, including a synergistic retrieval that we have developed for application to A-Train data.

### 3.3 Temperature and water vapor

In the previous sections, we find substantial increases in cloud occurrence and cloud ice above tropical cyclones. Previous
studies (Ueyama et al., 2018; Schoeberl et al., 2018) have suggested that injection of TTL cloud ice can lead to either hydration
or dehydration depending on the pre-existing conditions. Hence, it is important to examine the water vapor field above tropical
cyclones to ascertain both the sign and the magnitude of the (de)hydration impact.

    A joint AIRS-DARDAR retrieval method has been developed to retrieve atmospheric conditions above thick upper-tropospheric
clouds, combining hyperspectral infrared radiances from AIRS and collocated IWC profiles from DARDAR-Cloud. The re-
trieval method was described in detail and validated using a simulation experiment in Feng et al. (2021). Additional information
is provided in Appendix A to explain how the temperature and water vapor above tropical deep convection are retrieved.

    The AIRS FOVs selected for the synergistic retrieval are within $6.75$ km (half of the AIRS nadir footprint size) to the nearest
DARDAR cloud profile and have window band brightness temperatures ($BT_{1231}$) colder than 230 K. The $BT_{1231}$ threshold
ensures that liquid clouds can be neglected in the retrieval. The frequency of AIRS FOVs passing this criterion is shown in
Fig. 3 (i) and Fig. 4 (d). The selection of FOVs is illustrated in Fig. 6 for a tropical cyclone event. In this figure, the brightness
temperature in a window channel from AIRS L1B observation depicts a typical cyclonic cloud distribution. The vertical cross-
section illustrates the retrieval over selected AIRS FOVs. The same data selection and retrieval processes are performed for all

tropical cyclone overpasses. In total, 3475 FOVs from 345 tropical cyclone events are selected, with 2735 profiles successfully retrieved (reaching convergence in the iterative retrieval procedure). The converged retrievals are mostly located within 500 km of the cyclone center; their distributions are shown in Fig. 5 (b). 740 FOVs do not converge and are not used for further analysis. These FOVs typically have large radiance residuals between AIRS observations and the forward model calculations, where cloud states from DARDAR-Cloud and prior atmospheric states are used as inputs. Possible causes of non-convergences include 1) non-uniform cloud cover among the FOVs so that the $1.4 \, \mathrm{km} \times 1.8 \, \mathrm{km}$ CloudSat footprints are not representative of the $13.5 \, \mathrm{km} \times 13.5 \, \mathrm{km}$ AIRS FOVs, and 2) the optical thickness of the topmost cloud layer is smaller than one so that the assumption of vertically uniform optical properties per cloud mass does not hold in the radiative transfer calculations (Feng et al., 2021). The retrieved temperature, water vapor, and clouds, are shown as a function of vertical level and radial distance in Fig. 7 (a,e).

Owing to smaller horizontal sampling footprints and the availability of collocated cloud observations, the synergistic retrieval can reveal relatively small-scale variations in the thermodynamic fields above TTL clouds. To understand whether overshooting convection has a direct impact on water vapor, retrieved samples are classified into overshooting DCCs (DCC-OT), non-overshooting TTL clouds (TTL-OTHER), and non-TTL clouds (NTTL), using the same cloud classification introduced in Section 3.1 and Fig. 2. The converged profiles contain 731 DCC-OTs, 1508 TTL-OTHERs, and 496 NTTLs; the sample counts are shown in Fig. 5 (b). The mean profiles for each category are shown in Fig. 9 (a,b). These samples are representative of the cloud categories but may not be representative of the geographical pattern associated with tropical cyclone events because the number of samples within each radius bin is limited.

Meanwhile, similar cyclone-centered composites of thermodynamic fields are constructed using MLS v4.2 and ERA5 (Hersbach et al., 2020) in Fig. 7. The sample locations of ERA5 are identical to the measurement locations used for the joint AIRS-DARDAR retrieval. The measurement locations of MLS products used in this study are shown in Fig. 1 (c) and Fig. 5 (a). The sample density of MLS near the cyclone center is lower because only measurements not affected by high clouds are used. Figure 7 also shows composites of IWC from the synergistic retrieval (Fig. 7 (a,e)), from DARDAR-Cloud (Fig. 7 (b,f) corresponding to measurements shown in Fig. 1 (b)), and from ERA5 (Fig. 7 (c,g)) at retrieved sample locations. We note that there is no collocation between CloudSat/CALIPSO (DARDAR) and MLS in Fig. 7 (b,f) as only MLS observations not impacted by high clouds are selected.

The effects of cyclones on temperature and water vapor are examined by subtracting the multi-year monthly mean at every sample location from Fig. 7. The anomalous thermodynamic fields are presented in Fig. 8, while the mean anomalies for the three cloud categories (DCC-OT, TTL-OTHER, and NTTL) are shown in Fig. 9 (c,d). For MLS and ERA5, the monthly mean climatologies are constructed using the same dataset (MLS or ERA5, respectively). For the synergistic retrieval, there is no available 'climatology' from this retrieval dataset for non-cyclonic conditions. AIRS L2 temperature and MLS water vapor products are used instead, which are converted to the same vertical resolution to reduce systematic bias, as described in Appendix A.

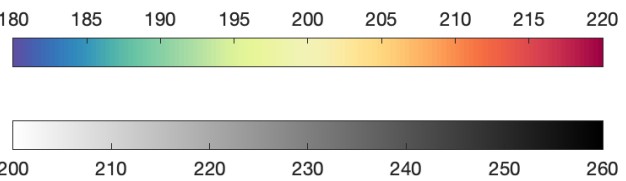

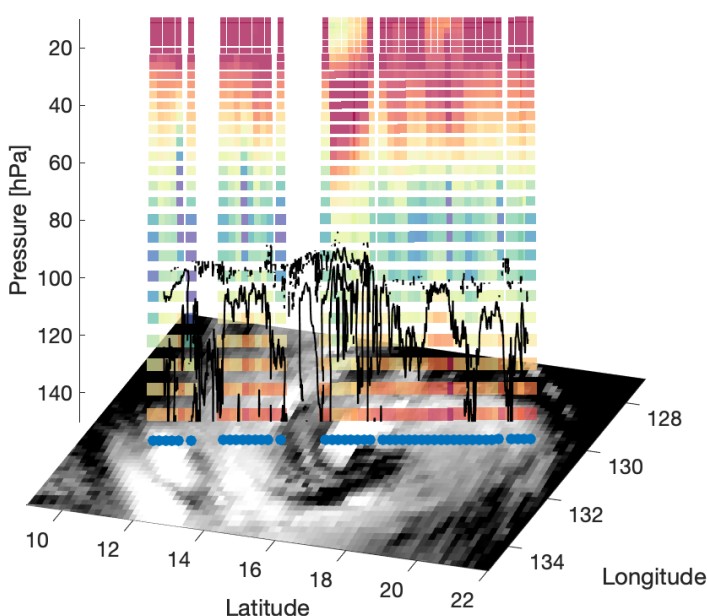

**Figure 6.** A-Train overpass and retrievals of a tropical cyclone event on October 2nd, 2007. The underlying image in greyscale shows the brightness temperature in a window channel ($BT_{1231}$ [K]) from the AIRS L1B product and indicates the cloud-top temperature. The vertical cross-section in color contours illustrates the temperature [K] retrieval over thick upper-tropospheric clouds ($BT_{1231}$ colder than 230 K) using the synergistic method described in the text; the black line marks the IWC at $10^{-4}$ g/m$^3$ based on the DARDAR data and outlines the cloud top positions.

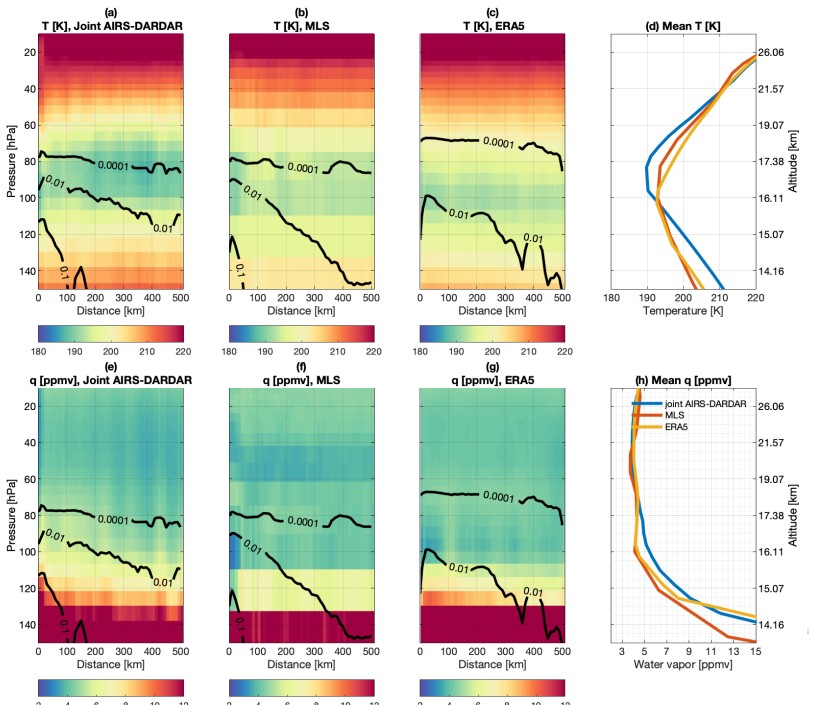

**Figure 7.** Cyclone-centered composites of temperature [shading, K], water vapor [shading, ppmv], and ice water content [black contour, $g/m^3$]. Temperature (a) and water vapor (e) from the joint AIRS-DARDAR retrieval; the sample counts are shown in Fig. 5 (b). Temperature (b) and water vapor (f) from the MLS v4.2 product; the sample counts are shown in Fig. 1 (c) and Fig. 5 (a). The IWC is from the DARDAR-Cloud; the sample density is shown in Fig. 1 (a). Temperature (c) and water vapor (g) from the ERA5 product sampled at the same locations as (a,e). Mean temperature (d) and water vapor (h) profiles from the different datasets.

### 3.3.1  Temperature

Using the synergistic retrieval, we find that tropical cyclone events lead to an oscillating pattern of temperature anomalies above the cloud top (Fig. 8 (a)). This pattern shifts the cold-point tropopause to higher altitudes (see also Fig. 9 (a)). Compared with the climatology, the mean temperature profile above cyclones shows a noticeable negative anomaly between 40 to 100 hPa and positive anomalies at other vertical ranges. We note that this cold anomaly around 80 hPa also supports the cold signature in $BT_{690}$ displayed in Fig. 3 (k). This vertically-oscillating anomaly feature is consistent with previous findings using radiosonde and GPS radio occultation measurements (Holloway and Neelin, 2007; Biondi et al., 2013; Rivoire et al., 2016).

The oscillating pattern of temperature anomalies may arise for a few reasons. The good alignment of the cold anomaly with the cloud top position around the cold point ($\sim$80 hPa) implies the impact of cloud radiative effects. This motivates us to ascertain the role of radiation in forming the retrieved temperature pattern. The thick cloud layers may absorb incoming solar radiation to heat the cloud top while attenuating emission from the warm surface to cool the atmospheric layers above the

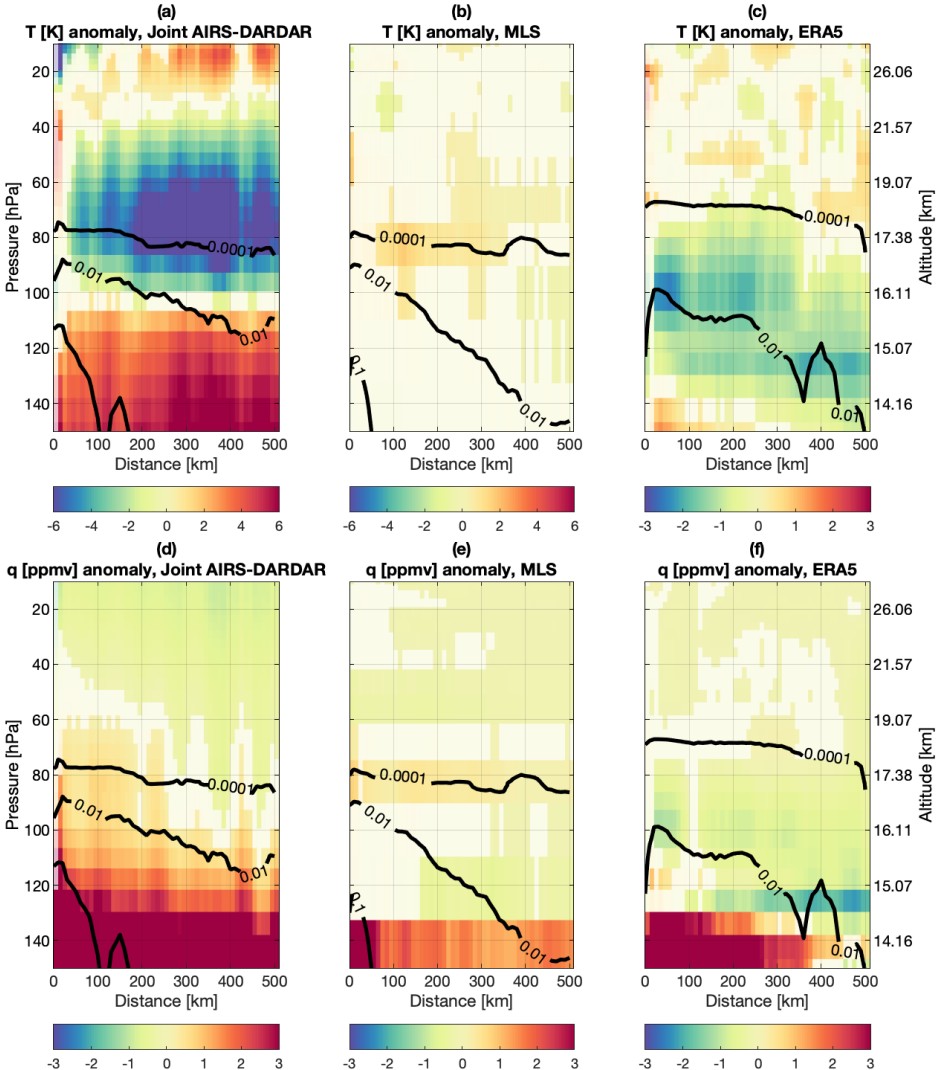

**Figure 8.** Cyclone-centered composites of temperature [shading, K], water vapor anomalies [shading, ppmv], and ice water content [black contour, $g/m^3$]. Anomalies below the 99% confidence level are set to be transparent. (a,d) Similar to Fig. 7 (a,e) but after subtracting the climatologies of temperature from the AIRS L2 product and water vapor from the MLS v4.2 product, respectively. (b,e) Similar to Fig. 7 (b,f) but after subtracting the climatologies of temperature and water vapor from the MLS v4.2 product. (c,f) Similar to Fig. 7 (c,g) but after subtracting the climatologies of temperature and water vapor from ERA5.

cloud. These expected cloud radiative effects agree with the signs of temperature anomalies and will be further examined in Section 5. On the other hand, Rivoire et al. (2020) pointed out that cloud radiative cooling only partly explains the cooling tendency above tropical cyclones. Other mechanisms at play may include the adiabatic expansion (Holloway and Neelin, 2007) in convective overshoots (Robinson and Sherwood, 2006) and the outward branches of secondary circulation (Rivoire et al., 2020; Schubert and McNoldy, 2010).

It is also interesting to find that the temperature anomaly is stronger above non-overshooting clouds (TTL-OTHERs) than over DCC-OTs or NTTLs (Fig. 9 (a,c)). This finding suggests a potential linkage between the temperature anomaly and the formation of TTL clouds. For example, one plausible explanation is that the cooling of air above cyclones promotes the formation of thicker TTL clouds by favoring water vapor deposition onto ice particles, as seen in Fig. 9 (d) and Fig. 10 and discussed in Section 3.3.2.

We note that the significant temperature anomaly pattern identified by the synergistic retrieval is not found in either MLS or ERA5. Livesey et al. (2017) have documented that MLS temperature retrievals are particularly susceptible to cloud contamination, while Schwartz et al. (2008) note that this issue cannot be effectively screened out by the $status$ flag of the MLS product. Therefore, the MLS temperature product may not be able to observe the pattern of temperature anomaly.

By comparing ERA5 (Fig. 7 (c) and Fig. 8 (c)) to the synergistic retrieval (Fig. 7 (a) and Fig. 8 (a)), we find that although ERA5 produces a higher cloud top (marked by $10^{-4}$ g/m$^3$ IWC contour) than the DARDAR observation, it generally underestimates TTL cloud ice. ERA5 also exhibits a cold anomaly but places it at lower altitudes compared to the synergistic retrieval, which is partly attributable to different radiative heating signatures due to differences in cloud ice. Previous studies (Wright et al., 2020) have also found large discrepancies in the upper-tropospheric temperature and tropical high clouds among reanalysis products, likely due to convective parameterization (Takahashi et al., 2016; Wright et al., 2020).

### 3.3.2 Water Vapor

Using the synergistic retrieval, Fig. 8 (d) shows that both hydration and dehydration can occur above cyclones. Hydration is found below 80 hPa, especially near the cyclone center, while dehydration is found above 60 hPa. This finding is consistent with MLS observations (Fig. 8 (e)), even though the synergistic retrievals are performed above thick upper-tropospheric clouds while the selected MLS observations are high-cloud free.

We note that the synergistic retrieval does not have sufficient vertical sensitivity to fully resolve the vertical distribution of water vapor, due to the smearing effect of the averaging kernel discussed in Appendix A. Nevertheless, the retrieval is sensitive to the spatial variability of the layer integrated water vapor above 16 km (LIWV), which has been validated by Feng et al. (2021) and in Appendix A. The retrieved LIWV above 16 km is shown in Fig. 10 (a, b) as a function of radial distance from the cyclone center. Therefore, we focus on the horizontal variability of LIWV above cyclones, which can be more confidently detected by the synergistic retrieval, to disclose overall hydration or dehydration above 16 km.

The synergistic retrieval detects a decreasing LIWV with increasing radial distance (Fig. 10 (a,b)). Significant hydration occurs near the cyclone center, increasing the LIWV by up to 0.18 g/m$^2$ ($\sim 9\%$), while dehydration occurs around a distance

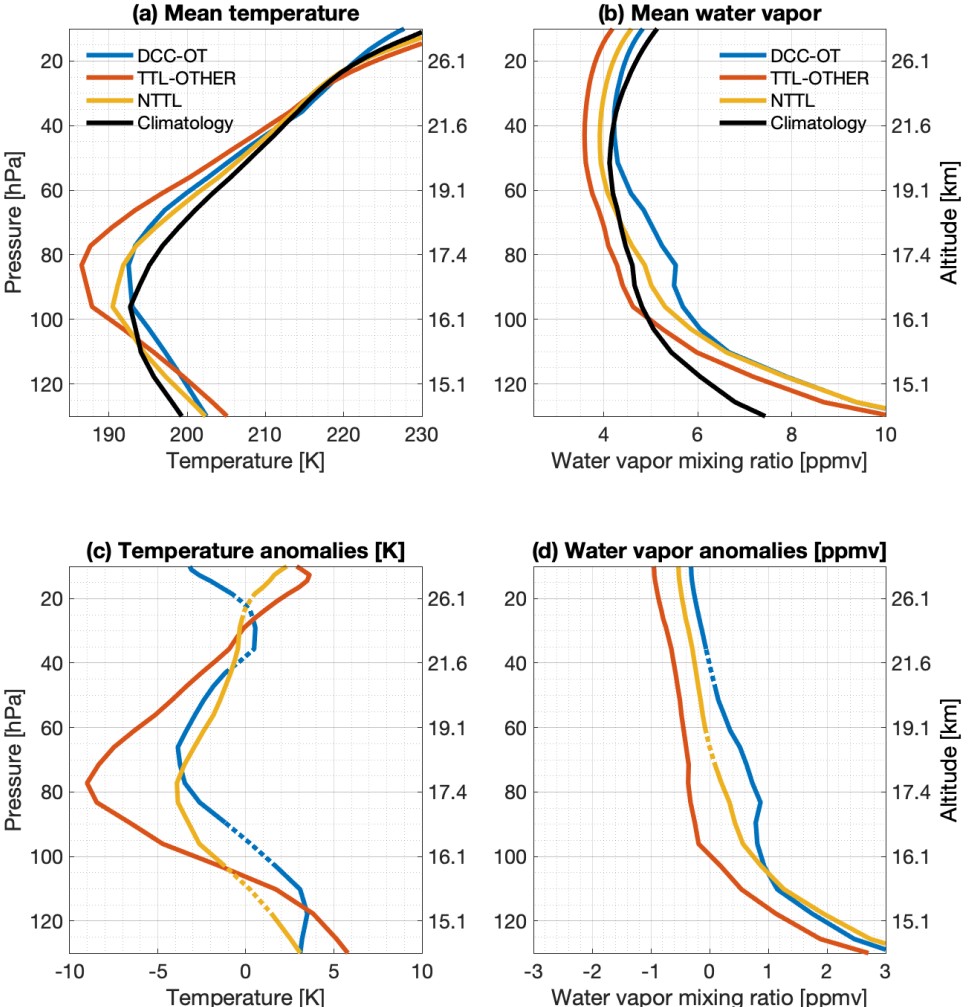

**Figure 9.** The mean (a) temperature [K] and (b) water vapor [ppmv] profiles above cyclones, for overshooting TTL clouds (DCC-OTs, blue), non-overshooting TTL clouds (TTL-OTHERs, orange), and non-TTL clouds (NTTL, yellow), along with the climatology (black). (c,d) the same as (a,b), but for anomalies with respect to the climatology.

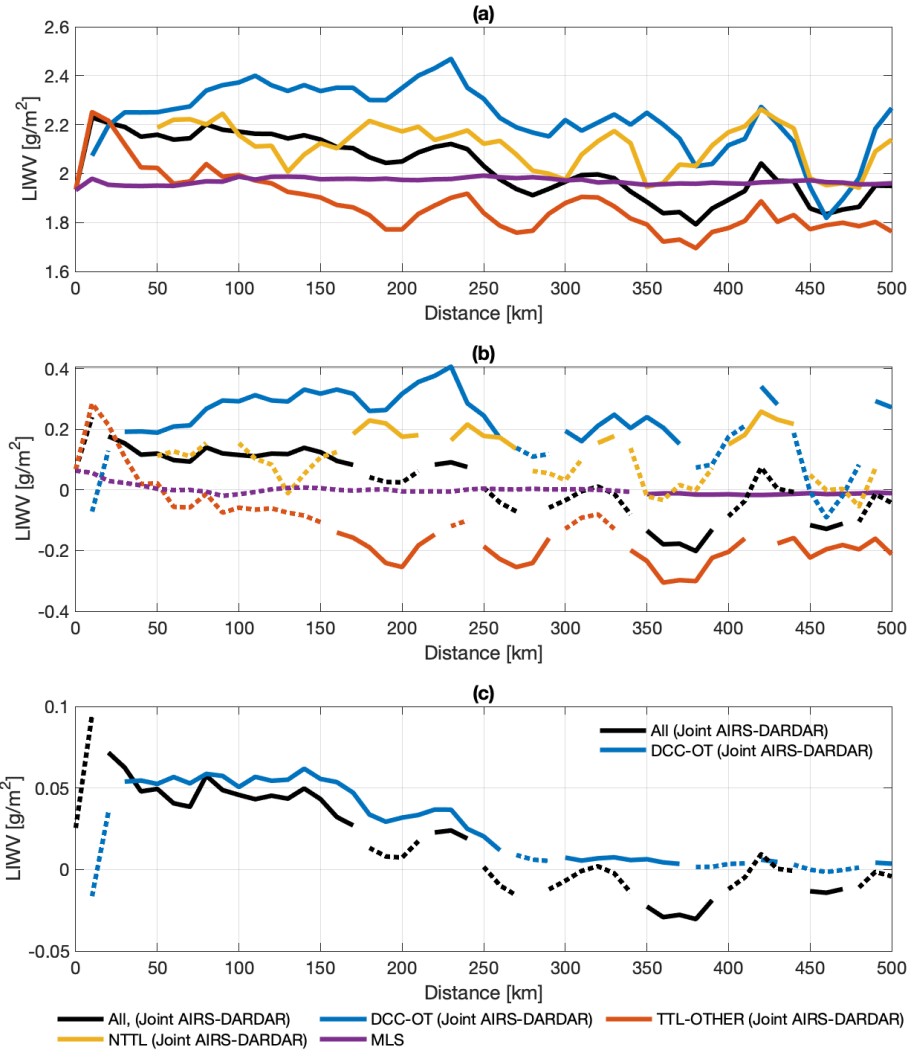

**Figure 10.** (a) Layer integrated water vapor above 16 km (LIWV) from the joint AIRS-DARDAR retrieval (black) and MLS (purple). Samples retrieved by the synergistic retrieval are separated into overshooting TTL clouds (DCC-OT, blue), other non-overshooting TTL clouds (red), and non-TTL clouds (yellow). Solid curves show the statistically significant (99% confidence level) anomalies while dashed curves are not statistically significant at the 99% level. (b) Same as (a) except for the anomaly in LIWV. (c) Contributions to the LIWV anomaly from thick upper-tropospheric clouds ($BT_{1231} < 230$ K, black) and DCC-OTs (blue).

of 375 km. In MLS, the LIWV does not substantially differ from the climatology, possibly because MLS samples large, high-cloud free areas.

    The (de)hydration impact of cyclones is classified after isolating DCC-OTs, TTL-OTHERs, and NTTLs. As depicted in Fig. 10 (a,b), significant dehydration is only found above non-overshooting TTL clouds, while significant hydration is found above overshooting clouds and non-TTL clouds. DCC-OTs increase the LIWV above 16 km by up to 0.4 $g/m^2$, which is equivalent
to 20 % of the climatological value. It suggests that overall the air above cyclones is hydrated by convection, especially overshooting convection that penetrates the base of the TTL. The non-overshooting TTL clouds (TTL-OTHER), however, are found to be associated with a drier environment, possibly due to the deposition of water vapor onto ice particles in colder temperatures (as suggested by Fig. 8 (a) and Fig. 9 (a,c)).

    The occurrence frequency of thick upper-tropospheric clouds (AIRS FOVs being colder than 230 K), above which the
355 synergistic retrieval is conducted, is shown in Fig. 4 (d). The expectation of changes in LIWV contributed by thick upper-tropospheric clouds is then estimated by multiplying the LIWV anomaly (Fig. 10 (b)) by the occurrence frequency of these clouds (Fig. 4 (d), blue and orange areas). Figure 10 (c) shows that the LIWV above cyclones is expected to be around 0.05 $g/m^2$ (1.5%) larger than the climatology within a 150 km radius of the cyclone center, and to be as much as 0.03 $g/m^2$ (0.9%) smaller at a distance of 375 km. On average, the stratospheric column above thick upper-tropospheric clouds within 500 km
of cyclone centers is 0.014 $g/m^2$ moister than the climatology. A similar calculation is performed for DCC-OTs, using the occurrence frequency of DCC-OTs shown by the blue area in Fig. 4 (b). It is found that DCC-OT alone increases the mean LIWV above tropical cyclones by 0.024 $g/m^2$, which is around 0.7 % of the climatological value.

    In summary, hydration is found to result from overshooting convection, which injects water substances directly. The moisture injected by overshoots hydrates the surrounding environment so that the cloud-free TTL (NTTLs) also shows higher LIWV
compared to the climatology. However, at locations away from overshoots, we find that overflow or pre-existing clouds in the TTL are associated with a colder and drier environment, potentially due to water vapor deposition onto ice particles.

## 4   Radiative effects

In the context of this paper, we have defined the lower boundary of the TTL to be the clear-sky level of zero-radiative heating (LZRH). Radiative heating rates in the TTL are crucial, for instance, to the cross-tropopause transport of water vapor, as they
potentially drive the diabatic ascent of air parcels across isentropic surfaces to the stratosphere. While the ascending motion that prevails in the tropical lower-stratosphere is driven by dissipating waves (Holton et al., 1995; Plumb, 2002), it is found that TTL clouds can heat the air through infrared radiation and is important to explain the mass flux to the stratosphere (Corti et al., 2006; Yang et al., 2010). Hence, we are interested in how radiative heating rates are perturbed above tropical cyclones and whether this helps retain the moisture anomaly in the TTL.
Moreover, as indicated by the close alignment of cloud boundary and temperature anomalies, cloud radiative effects may have played a role in forming the temperature anomalies seen in Fig. 8. Therefore, a cyclone-centered composite of radiative heating rates is constructed using the CloudSat 2B-FLXHR-LIDAR product to help address these questions.

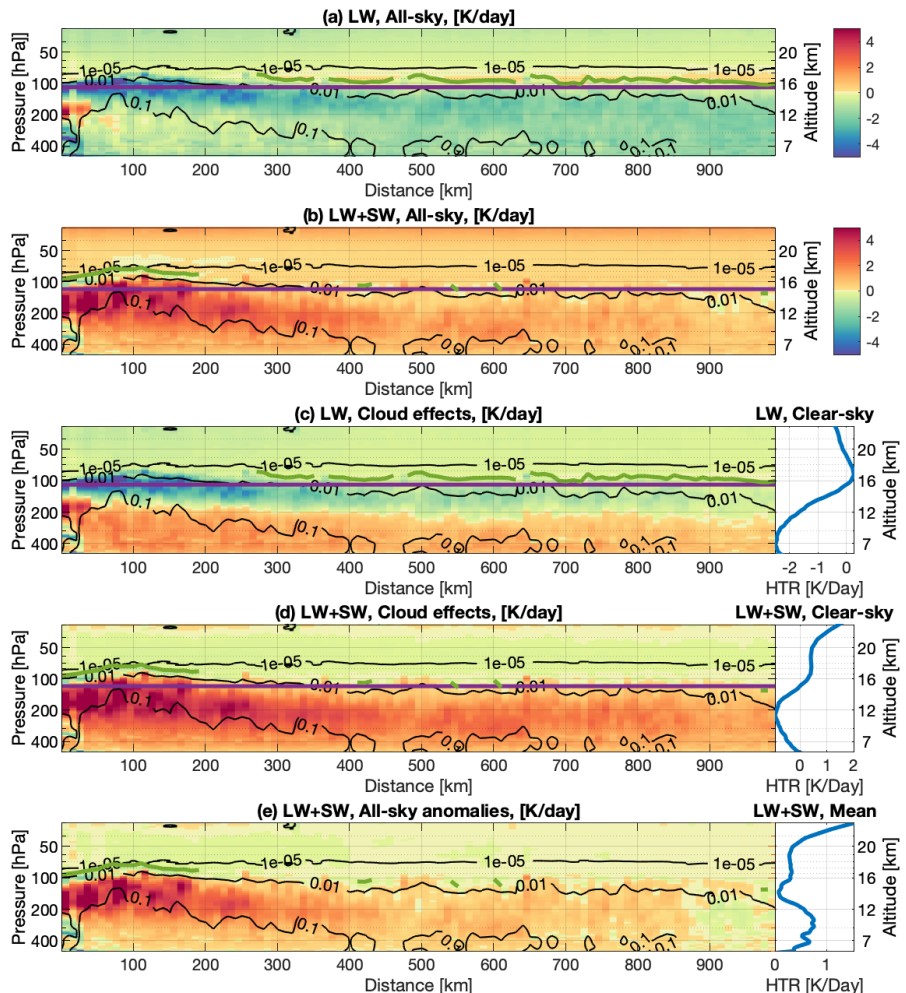

**Figure 11.** Radiative heating (shading, K/day, from CloudSat 2B-FLXHR-LIDAR) and IWC (black contour, g/m³, from DARDAR-Cloud) as a function of distance to cyclone center. (a) All-sky longwave and (b) net (longwave + shortwave) radiative heating rates. (c,d) Same as (a,b) but for cloud radiative effects, which are defined as the differences between all-sky heating rates and the mean clear-sky heating rate (blue curve in the right panel). (e) Same as (d) but for all-sky net radiative heating anomalies, which are defined as the differences between all-sky heating rates and the mean all-sky heating rate. The Magenta line marks the clear-sky LZRH. The green line marks the cloudy-sky LZRH, determined as the vertical position where the heating rate changes from positive to negative.

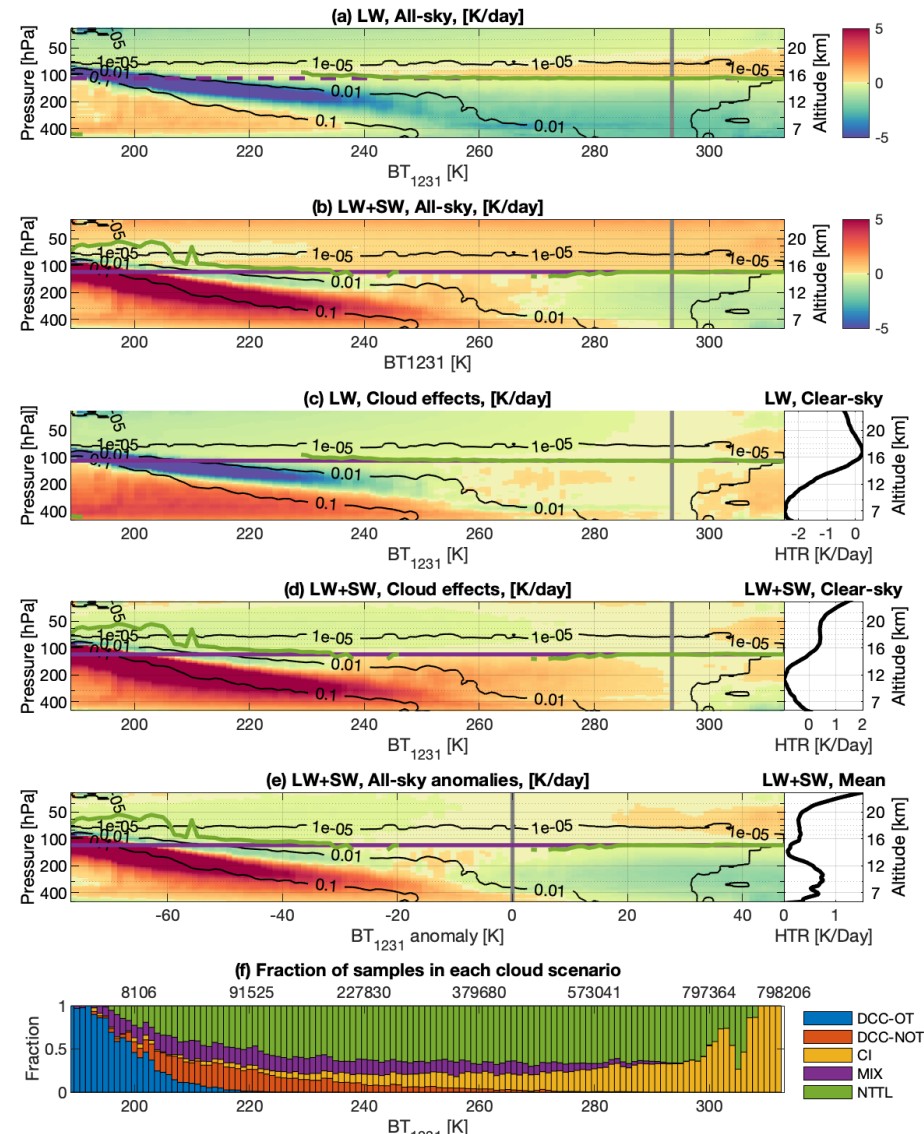

**Figure 12.** Radiative heating rates (shading, $K/day$, from CloudSat 2B-FLXHR-LIDAR) and IWC (black contour, $g/m^3$, from DARDAR-Cloud) as a function of window band radiance $BT_{1231}$ (K, from collocated AIRS L1B). (a) All-sky longwave and (b) net (longwave + shortwave) radiative heating rates. (c,d) Same as (a,b) but for cloud radiative effects, which are defined as the differences between all-sky heating rates and the mean clear-sky heating rate (black curve in the right panel). (e) Same as (d) but for all-sky net radiative heating anomalies, which are defined as the differences between all-sky and the mean all-sky cyclone overpass (black curve in the right panel). (f) The proportion of samples in each cloud category (classified in Section 3.1) to all cloudy overpass samples. The numbers on the top indicate the number of cloudy samples with $BT_{1231}$ colder than the corresponding temperature marked at the bottom. The Magenta line marks the clear-sky LZRH. The green line marks the cloudy-sky LZRH, determined as the vertical position where the heating rate changes from positive to negative.

The cloud radiative effect, measured by the radiative heating rate difference between all-sky and clear-sky overpasses, is shown as a function of pressure level and radial distance from the cyclone center in Fig. 11 (c,d). Only daytime samples (overpasses at 13:30 local solar time) are used to exclude the lack of shortwave heating during the nighttime. We identify the LZRH in the heating rate profiles as the level where heating rates change from negative to positive. Note that there is no LZRH identified in two conditions: in Fig. 11 (a,c) when the longwave heating rate is entirely negative in the TTL, and in Fig. 11 (b,d,e) when the net heating rate is entirely positive.

In the longwave, Fig. 11 (c) shows that clouds generally produce positive perturbations inside the clouds (below 200 hPa) but strong cooling at the atmosphere near and above the cloud top (marked by the 0.01 $g/m^3$ IWC contour). Figure 11 (d) shows that this cloud top longwave cooling effect is offset by the shortwave effect. The compensation between longwave and shortwave leads to a net cloud radiative cooling effect in the layers above the cloud top but a net cloud radiative heating effect below the cloud top. These effects shift the clear-sky LZRH to higher altitudes, suppressing diabatic ascent in the TTL.

The cloud radiative heating composited here, particularly the in-cloud heating (below 0.01 $g/m^3$ IWC contour) and cloud-top cooling feature (above 0.01 $g/m^3$ IWC contour), corroborates the temperature structure retrieved using the synergistic retrieval method (Fig. 8 (a)). It is also consistent with the finding of Rivoire et al. (2020) based on an analysis of COSMIC data, who noted a cooling tendency above 100 hPa that is partially attributable to radiative effects.

Fig. 11 demonstrates that the qualitative structure of cloud radiative heating/cooling effects change little with radial distance from the cyclone center. This insensitivity, however, hides the distinct radiative effects of different types of clouds. For instance, it is well known that DCC cools the atmospheric column above it by trapping the thermal fluxes within the cloud, while thin cirrus can warm the air locally by absorbing the thermal emission from the surface (e.g., Fig. 5 (b) of Rivoire et al. (2020)). The compensating sign of radiative effects is not evident anywhere in Fig. 11 (c), despite the fact that cloud types have a strong radial dependence (Fig. 4 (b)). Recognizing that the cloud radiative effects are determined by the cloud optical depth, $BT_{1231}$ can be used to characterize the radiative effects of different types of clouds owing to its sensitivity to cloud optical depth over tropical ocean FOVs.

To construct a composite of cloud radiative heating with respect to $BT_{1231}$, cloud profiles at CloudSat footprints are paired with the nearest AIRS spectra. The AIRS FOVs evaluated here are limited to those with scanning angles less than 14 degrees, which has a negligible (<3%) effect on the optical depth and those with CloudSat samples locations fall within their ground footprints (13.5 km). A composite of all-sky radiative heating rates over every 1-K bin of $BT_{1231}$ is shown in Fig. 12, along with differences relative to clear sky conditions.

Fig. 12 shows that different regimes of cloud radiative effects are well differentiated by $BT_{1231}$. When $BT_{1231}$ is colder than 230 K (indicating thick upper tropospheric clouds, as discussed in Section 3), net radiative cooling is observed in the TTL. This net cooling is largely caused by longwave cooling above DCC cloud tops (indicated by IWC contours in Fig. 12). Note that this BT condition (colder than 230 K) occurs in more than 50 % of overpasses within a 300 km radius of the cyclone center (Fig. 4 (d)), indicating that the TTL is dominated by radiative cooling within this range. In this cloud regime, the cloud effect lifts the LZRH to higher altitudes, reaching 19.5 km when $BT_{1231}$ is around 200 K. When $BT_{1231}$ is between 240 K and 280 K, noticeable heating emerges near the cloud top (compare Fig. 12 (a) and (b)), attributable to the deeper penetration of solar

radiation into the less opaque cloud layer. When $BT_{1231}$ is greater than 294 K (the clear-sky climatological mean value), TTL heating is evident. This TTL heating is accompanied by a substantial increase of CI (cirrus) scenes with increasing $BT_{1231}$ as shown in Fig. 12 (f), which suggests the contribution of thin cirrus in the TTL heating as mentioned above.

Furthermore, we compute the net heating anomaly with respect to the all-sky climatology. The all-sky net heating anomaly is then shown as a function of the $BT_{1231}$ anomaly, which is also defined with respect to the all-sky average, in Fig. 12 (e). It is clear that cloud effects on TTL heating rates above 16 km are well-differentiated by $BT_{1231}$: cooling when the $BT_{1231}$ anomaly is negative and heating when the $BT_{1231}$ anomaly is positive. Given that a negative $BT_{1231}$ anomaly prevails within 1000 km of the cyclone center (see the black line in Fig. 4 (d)), it is no wonder that Fig. 11 generally shows TTL cooling above tropical cyclones.

The prevalence of TTL radiative cooling in Fig.12 (c,d) suggests that the diabatic ascent that normally (climatologically) occurs within the TTL is greatly suppressed by cloud radiative effects in a 2000 km × 2000 km domain surrounding tropical cyclone centers. A hydrated air parcel above a cyclone has to be advected further away from the cyclone centers to experience radiative heating and ascend to the stratosphere.

Finally, it is worth noting a few caveats of the cloud radiative effect assessed here. As the cloud radiative effect is measured by the difference in radiative heating between all-sky and clear-sky conditions, the result is subject to differences in surface emissions and thermodynamic conditions between the clear-sky and all-sky situations. We cannot quantify how much of the TTL cooling (as shown in Fig. 12 (d,e)) is directly attributable to clouds because large anomalies in temperature and water vapor (as shown in Fig. 8) also exist above tropical cyclones. It is unclear how much these non-cloud variables account for the radiative heating anomalies shown in Fig. 11 and 12. Moreover, the CloudSat radiative heating data used here may be subject to errors because their calculation is based on the ECMWF forecast which does not fully capture the above-cyclone temperature and water vapor perturbations (see Fig. 8). It is therefore useful to examine the heating rate change above these tropical cyclones using collocated observations of cloud, temperature, and water vapor profiles from our synergistic retrieval.

## 4.1 Heating rate decomposition

Large temperature and water vapor anomalies in the TTL above tropical cyclones (as depicted in Fig. 8) are detected using the joint AIRS-DARDAR retrieval method. Here, using the radiative transfer model RRTM (Iacono et al., 2000), the radiative effects of the cloud, temperature, and water vapor anomalies are isolated in Fig. 13. The shortwave effects of temperature and water vapor are not shown because they are negligible compared to the longwave effects.

Following the Partial Radiative Perturbation approach (Wetherald and Manabe, 1988), we measure the radiative effect of a variable by differencing the RRTM computations with perturbed and unperturbed values of this variable. For instance, the radiative effect of cyclonic clouds is measured as:

$$dHTR(c) = HTR(c, t0, q0) - HTR(c0, t0, q0) \tag{1}$$

Here, $HTR$ denotes the instantaneous heating rate profile and is computed using RRTM, and $c$, $t$, and $q$ denote cloud, temperature, and water vapor profiles from the cyclone samples, respectively. Note that for the $t$ and $q$ profiles, only the

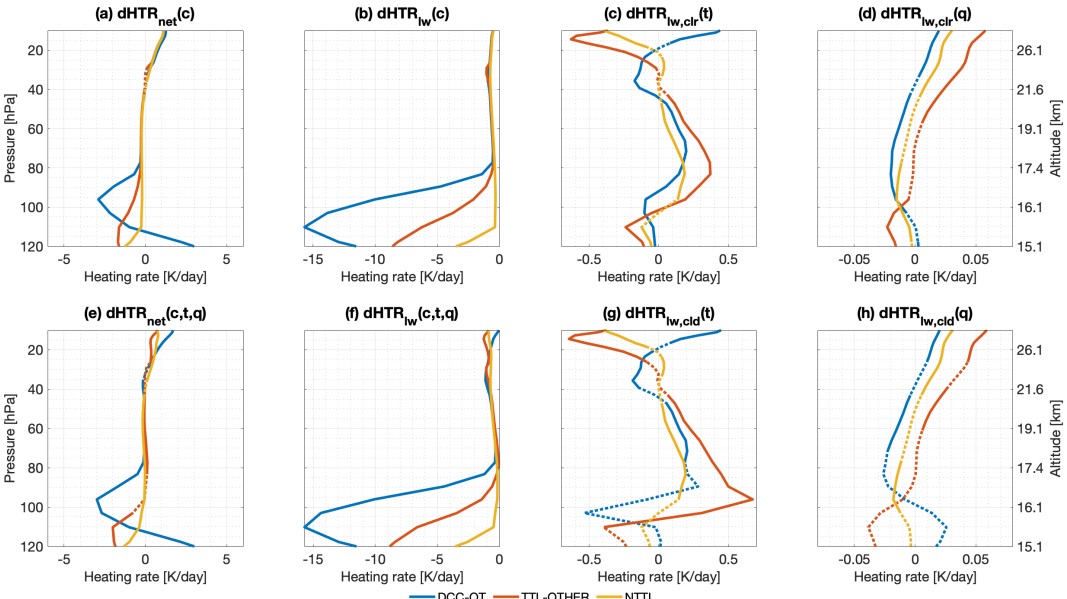

**Figure 13.** The radiative heating effects of cloud, temperature, and water vapor. (a) The net effects of cloud, $dHTR_{net}(c)$. (b) The longwave effects of cloud, $dHTR_{lw}(c)$. (c) The longwave effects of temperature for clear-sky conditions, $dHTR_{lw,clr}(t)$. (d) The longwave effects of water vapor for clear-sky conditions, $dHTR_{lw,clr}(q)$. (e) The net effects of cloud, temperature and water vapor, collectively, $dHTR_{net}(c,t,q)$. (f) The longwave effects of cloud, temperature, and water vapor, collectively, $dHTR_{lw}(c,t,q)$). (g) The longwave effects of temperature for cloudy-sky conditions, $dHTR_{lw,cld}(t)$. (h) The longwave effects of water vapor for cloudy-sky conditions, $dHTR_{lw,cld}(q)$.

portions above 16 km are of concern and replaced in the PRP computation. Variables with subscript 0 denote values from the all-sky climatology. The mean instantaneous longwave and net radiative heating rate profiles for DCC-OTs, TTL-OTHERs, and NTTLs are shown in Fig. 13. The radiative effects of temperature and water vapor are examined in both all-sky conditions, denoted by the subscript $cld$ in Fig. 13 (g,h), and clear-sky conditions ($c = 0$), denoted by the subscript $clr$ in Fig. 13 (c,d).

By comparing Fig. 13 (a) and (e), we find that the total net radiative effect, $dHTR_{net}(c,t,q)$, is dominated by clouds. The sign and magnitude of the cloud radiative effect are consistent with our previous conclusion, namely a cooling effect at the cloud top as indicated by a cold $BT_{1231}$ anomaly (Fig. 12 (a,e)). The cloud longwave cooling effect around 90 hPa is much larger above DCC-OTs due to higher cloud ice water content near this level.

As seen in Fig. 13 (g,h), above 80 hPa where cloud ice diminishes, the all-sky radiative effects of anomalies in temperature
and humidity become more important. Temperature modulate longwave emission in ways that damp the temperature anomalies (compare Fig. 13 (g) to Fig. 9 (c)). For water vapor, a moistening at the cold point increases thermal emissivity, which leads to cooling at the cold point and heating at lower levels. Therefore, TTL hydration above DCC-OTs and NTTLs leads to radiative cooling in the TTL. Assuming a similar pattern in thermodynamic anomalies to what is shown in Fig. 9 (c,d) in the clear-sky,

we compute the radiative effects of temperature and water vapor above 80 hPa in the clear-sky (Fig. 13 (c,d)) and find that
they are similar to the all-sky results.

Despite the limited vertical resolution of the water vapor retrieval, as discussed in Appendix A, it is unambiguous that TTL
hydration leads to radiative cooling. It suggests that moisture above overshooting convection radiatively cools the layer, thus
constraining the moist air from diabatically ascending to higher altitudes. This finding is consistent with the robust radiative
cooling seen above cyclones in Fig. 12.

## 5 Conclusions

In this study, we aim to understand the impacts of tropical cyclones on thermodynamic conditions in the TTL using multiple
instruments aboard the A-Train satellites. We use a TC-overpass product to composite multiple observation products relative
to 947 tropical cyclone center locations over the northern part of the West-Pacific region to ascertain the effect of cyclones. To
address the lack of reliable observations of temperature and water vapor when thick convective clouds are present, a retrieval
scheme proposed by Feng and Huang (2018) is improved by incorporating cloud properties measured by active sensors to
retrieve the above-cyclone temperature and water vapor profiles simultaneously.

This study finds that tropical cyclones substantially increase TTL clouds. These TTL clouds occur frequently above tropical
cyclones, mostly as non-convective high clouds above a clear troposphere (CI category, Fig. 4 (b)). This distribution of high
clouds is consistent with in-situ aircraft observations (e.g., Jensen et al., 2013). Our finding emphasizes that the occurrence of
TTL cloud is 0.372 on average (Fig. 3 (a) and Fig.3(a)) within the 2000 km × 2000 km cyclone-centered composite domain,
highlighting the importance of tropical cyclones in generating TTL clouds. In contrast to the horizontally extensive occurrence
of TTL cloud cover, TTL cloud ice is most concentrated near the cyclone center (Fig. 3 (c) and Fig. 4 (a)), as a result of direct
convective overshooting and detrainment (Fig. 4 (c)). Furthermore, we find that the northwestern quadrant of the composite
domain is less impacted by cyclones Fig. 3 (b,j). There is also a persistent southwest preference in TTL cloud ice and DCCs.

Both MLS v4.2 and the joint AIRS-DARDAR method are used to evaluate the impact of tropical cyclones on temperature
and water vapor by compositing these fields as a function of radial distances to cyclone centers (Fig. 7). Differences between
the products are noticed in this study and they can arise for several reasons. First, contamination of cloud may cause artifacts in
MLS v4.2 product, especially in temperature, that may not be effectively screened out by the cloud flag of the product (Schwartz
et al., 2008; Livesey et al., 2017). Second, sampling differences (MLS is off-nadir while AIRS, CPR, and CALIOP are nadir-
looking) caused by instrument viewing geometry and the selection of samples can be substantial (see Fig. 5). Due to issues
caused by cloud contamination, any MLS FOVs containing high cloud are excluded in this study for both temperature and
water vapor. The implication of the joint AIRS-DARDAR method, however, exclusively selects FOVs that contain thick high
clouds to perform the retrieval. Hence, the thermodynamic conditions of samples with or without high clouds can lead to large
differences in the constructed cyclone composites. Third, the horizontal resolution of the joint AIRS-DARDAR method is at
least ten times finer than MLS. As a result, the joint AIRS-DARDAR method better captures the horizontal distribution pattern

(Feng et al., 2021). Last, compared to MLS, the joint AIRS-DARDAR method has higher precision and vertical resolution for temperature but lower precision and vertical resolution for water vapor.

Hence, the joint AIRS-DARDAR method is more advantageous in evaluating temperature fields related to tropical cyclone events. Figure 8 shows a noticeable pattern of vertically-oscillating temperature anomalies above tropical cyclones that lifts the cold-point tropopause level to higher altitudes. After investigating the cloud radiative effects in Section 4, we find that the signs of the temperature anomaly agree well with cloud effects on radiative heating rates, for example, the in-cloud warming (below the 0.01 $g/m^3$ IWC contour in Fig. 8 (a) and Fig. 12 (d)) due to shortwave heating and cloud-top cooling due to longwave cooling. Environmental cooling above clouds may also facilitate the formation of TTL clouds that deplete moisture from the detrainment of tropical cyclones, as indicated by the drier TTL over non-overshooting clouds. The cooling effect of tropical cyclones on cold-point temperatures also implies the importance of deep convection in modulating the stratospheric water vapor, so that a strong linkage between stratospheric water vapor and cold-point temperature, as noted by Randel and Park (2019), cannot preclude the role of convection in water vapor variability.

The joint AIRS-DARDAR method is also used to investigate overall (de)hydration above overshooting deep convective clouds and other non-overshooting clouds. Figure 8 (b) suggests that cyclones mostly hydrate the atmospheric column above them. Above overshooting deep convective clouds, the LIWV is found to be 40 % higher than the local climatology (Fig. 10 (b)). Substantial hydration is also found above clouds located beneath 16 km (NTTL). We suspect that this is likely from advected moist plumes from overshooting injection, though we are unable to prove these suspicions at this time. After isolating different cloud categories, dehydration is only found above non-overshooting TTL clouds (TTL-OTHERs) which are coincidently associated with colder temperatures than other cloud categories (Fig. 9 and 10). The coexistence of dehydration (as opposed to the moistening above other cloud categories), cold anomalies, and non-overshooting TTL clouds suggests that in this situation water vapor is likely deposited onto ice particles.

By comparing the nearest thermodynamic profiles from the ERA5 reanalysis (Fig. 8 (c,f)) to the synergistic retrieval (Fig. 8 (a,d)), we find that the cold anomaly in ERA5 is at a lower altitude, which is partially attributable to biases in cloud ice in the reanalysis. The moistening signals around 80 hPa, as detected by the synergistic retrieval and MLS (Fig. 8 (d,e)), are not shown in ERA5. These results suggest that the above-cyclone water vapor in reanalysis products may be susceptible to the ability of models to simulate cloud ice and temperature using convective parameterizations (Wright et al., 2020).

Furthermore, we find that the cloud radiative effect is well-differentiated by $BT_{1231}$. Clouds heat the TTL via radiation when $BT_{1231}$ shows a warm anomaly and cool the TTL when $BT_{1231}$ shows a cold anomaly. Radiative cooling prevails above DCCs and thick anvils, which greatly reduce $BT_{1231}$. Radiative warming becomes more noticeable away from the cyclone center over thin cirrus. The radiative cooling anomaly further impacts the diabatic heating budget above tropical cyclones, suppressing diabatic ascent and air mass transport across isentropic surfaces to higher altitudes. It remains unclear how this suppressed diabatic ascent, together with the strong horizontal divergence created by the pressure gradient above the cyclone, affects stratosphere-tropopause exchange and the water vapor budget. To elucidate this effect in the trajectory modeling in future work, it will require the use of instantaneous radiative heating computed from deep convection-perturbed TTL thermodynamic

conditions (temperature, humidity, and cloud), as opposed to climatologic or reanalysis heating profiles that do not properly represent the convective perturbations.

Finally, we would like to highlight the advantages of the synergistic method in retrieving the above-cloud thermodynamic conditions. This method takes advantage of collocated infrared hyperspectra and active sensors and is capable of retrieving temperature and water vapor for overcast cloud conditions. These features are highly complementary to other datasets, including
530 MLS v4.2 and AIRS L2 v6, that are limited to clear-sky conditions. So far, this approach has only been applied to limited samples in the vicinity of tropical cyclones. It can be applied to other tropical and extra-tropical convective events, with potential implementation in other hyperspectral infrared sounders, such as IASI (Infrared Atmospheric Sounding Interferometer), CrIS (Cross-track Infrared Sounder), IRS (Infrared Spectrometer), and GIIRS (Geostationary Interferometric Infrared Sounder), to provide thermodynamic information over deep convective clouds on a global scale in future research.

*Author contributions.* YH conceived the cloud-assisted retrieval idea; JF implemented this idea with improvements using A-Train measurements. JF and YH co-designed the study of the tropical cyclone impacts and collectively wrote this paper.

*Competing interests.* The authors declare that they have no conflict of interest.

*Acknowledgements.* We thank Jonathon Wright, Xun Wang, Kevin Bloxam, Lei Liu, Louis Rivoire, and two anonymous reviewers for their constructive comments. This work is supported by grants from the Canadian Space Agency (16SUASURDC and 21SUASATHC)
and the Natural Sciences and Engineering Research Council of Canada (RGPIN-2019-04511). JF acknowledges the support of a Milton Leong Graduate Fellowship of McGill University. We thank Natalie Tourville for making the TC overpass dataset publically accessible ( https://adelaide.cira.colostate.edu/tc/ ). We thank ICARE Data and Services Center ( http://www.icare-lille1.fr ) and Dr. Julien Delanoë for access to the DARDAR product.

## Appendix A: Joint AIRS-DARDAR retrieval algorithm

Feng and Huang (2018) applied a cloud-assisted retrieval to AIRS L1B infrared radiance from FOVs filled with deep convective clouds, assuming a blackbody cloud top. This method retrieves atmospheric temperature and humidity profiles above the cloud top as described in Section 2.1 and is especially advantageous for overcast conditions during tropical cyclone events. An updated version of this retrieval method has been validated by Feng et al. (2021) using simulation experiments and is adopted here. We briefly describe this retrieval method below and interested readers can find more details of this retrieval method from
Feng et al. (2021).

In this method, we retrieve atmospheric states $x$ that include temperature, humidity, ice water content (IWC), and effective radius as an optimal estimation (Rodgers, 2000) that combines the $a\ priori$ of $x$ and the observation vector, $y$. The relationship

between the state vector and the observation vector is expressed as:

$$y = F(x_0) + \frac{\partial F}{\partial x}(x - x_0) + \varepsilon$$

$$= y_0 + K(x - x_0) + \varepsilon \tag{A1}$$

Where $F$ is the forward model, $K$ is the Jacobian matrix, which is a first-order linear approximation of $F$, and $\varepsilon$ is the residual. $x_0$ is the first guess, for which we use the mean of the *a priori*. Following the synergistic retrieval method (Feng et al., 2021), additional observation vectors are added, so that $y$ is formed as $[y_{rad}, y_{iwc}, y_{R_e}, y_{atm}]$. $y_{rad}$ is the infrared hyperspectra from the AIRS L1B product, for which 1109 channels are selected, based on the radiometric quality of each channel. This rigorous channel selection also excludes O3 absorption channels (980-1140 $cm^{-1}$), CH4 absorption channels (1255-1355 $cm^{-1}$), and shortwave infrared channels (2400-2800 $cm^{-1}$). $y_{iwc}$ is a 2-km vertical IWC profile from the DARDAR-Cloud product, from 1.5 km below the DARDAR-identified cloud top to 0.5 km above it. $y_{R_e}$ is the effective radius, which holds constant through vertical layers of an atmospheric column. $y_{atm}$ is the temperature and humidity profile from the nearest ERA5 reanalysis product (hourly, $0.25 \times 0.25$). The *a priori* dataset is obtained from the AIRS L2 supplementary product for temperature and water vapor and from the DARDAR-Cloud product for IWC and effective radius using data from 2006 to 2014 in the Northern part of the West Pacific.

The forward model to convert the atmospheric states to $y_{rad}$ is a radiative transfer model, the line-by-line version of MODerate spectral resolution TRANsmittance (MODTRAN 6.0 Berk et al., 2014). Following Feng et al. (2021), user-defined extinction coefficients, single-scattering albedo, and asymmetry factor are input to MODTRAN, assuming vertically uniform optical properties (per unit ice mass) and solid-column crystal habit. Forward model uncertainties caused by this assumption is evaluated in Feng et al. (2021) and is around 0.1 K in the mid-infrared. To relate $x$ to $y_{iwc}$ and $y_{atm}$, the forward model and the corresponding Jacobian matrix work as a linear interpolation matrix (Bowman et al., 2006; Feng et al., 2021).

Using the Gaussian-Newton method, the best estimate of $x$, $\hat{x}$ , is iteratively solved as:

$$\hat{x}_{i+1} = x_0 + (K_i^T S_\varepsilon^{-1} K_i + S_a^{-1})^{-1} K_i^T S_\varepsilon^{-1}[y - F(\hat{x}_i) + K_i(\hat{x}_i - x_0)]$$

$$\hat{x}_{i+1} = x_0 + (K_i^T S_\varepsilon^{-1} K_i + S_a^{-1})^{-1} K_i^T S_\varepsilon^{-1}[y - F(\hat{x}_i)] + A(\hat{x}_i - x_0) \tag{A2}$$

Where the subscript $i$ refers to the ith time step. $S_a$ and $S_\varepsilon$ are the covariance matrix of the *a priori* dataset and the observation vector, respectively, which are constructed the same as in Feng et al. (2021). More specifically, $S_\varepsilon$ is a diagonal matrix. The diagonal elements of $S_\varepsilon$ for $y_{rad}$ contain the sum of the square of radiometric noise of the instrument and the forward model uncertainty. For $y_{iwc}$, they contain the square of a doubling of posterior uncertainty of the IWC profile which is provided by the DARDAR-Cloud product at every measurement location and vertical level. For $y_{R_e}$, they contain the square of a doubling of posterior uncertainty of the effective radius of a vertical layer where optical depth measured from the cloud top reaches unity (Feng et al., 2021). The $S_\varepsilon$ for $y_{atm}$ is denoted as $S_{atm}$; it contains the square of a doubling of the root-mean-square difference between collocated ERA5 and MLS v4.2 products at every vertical level of ERA5. These uncertainty estimations are amplified in order to account for uncertainties caused by differences in the size and location of FOVs of different instruments.

The $A$ in Eq. A3 is referred to as averaging kernel. Given a 'truth' state vector that $F(x_t) = y$, it links the truth state to the retrieved state, so that:

$$\hat{x} - x_0 = A(x_t - x_0).$$ (A3)

Therefore, it regulates the vertical shape of the posterior estimation.

The iteration converges when:

$$(\hat{x}_{i+1} - \hat{x}_i)^T S(\hat{x}_{i+1} - \hat{x}_i) \ll N,$$ (A4)

where $N$ is the dimension of the state vector, and S is the posterior covariance matrix, which is computed combining the covariance matrix of $a\ priori$ and observation vector:

$$S = (S_a^{-1} + K^T S_\varepsilon^{-1} K)^{-1}$$ (A5)

Therefore, the retrieved state converges to $\hat{x}$ with an uncertainty range equivalent to the square root of the diagonal element
of the posterior covariance matrix $S$.

Feng et al. (2021) used a simulation experiment to evaluate a synergistic retrieval approach that combines infrared spectra ($y_{rad}$) with another observation vector that includes the IWC product from collocated observation vector, denoted as $y_{iwc}$, and additional atmospheric observations, denoted as $y_{atm}$. In this simulation experiment, the 'truth' atmospheric condition is simulated from a cloud-resolving model during a tropical cyclone event. We mimic the infrared spectra from AIRS observation
by adding synthetic noise to the forward model simulated infrared radiances that follow the spectral response function of AIRS. The IWC observation is simulated by perturbing the 'truth' IWC profile following the mean posterior uncertainty range provided by the DARDAR-Cloud product within 1000 km from cyclone centers. Therefore, the simulation experiment is designed to evaluate the realistic retrieval performance above thick upper-tropospheric clouds using AIRS L1B and DARDAR-Cloud products, with the same $S_\varepsilon$ for the observation vectors.

The synergistic retrieval performed here is similar to Case 4 in Feng et al. (2021); the only differences are in the $a\ priori$ dataset and the $y_{atm}$, which Feng et al. (2021) constructed hypothetically using numerical model simulation. To examine the capacity of the retrieval method in revealing the realistic thermodynamic conditions, a simulation experiment similar to Feng et al. (2021) is performed here, using the $a\ priori$ dataset and $S_\varepsilon$ we introduced earlier. Figure A1 shows the horizontal distribution of temperature and water vapor at 81 hPa, as well as the integrated water vapor above 16 km from the simulated
'truth', the prior, the nearest ERA5 ($y_{atm}$), and the posterior of the retrieval. In Fig. A1, the posterior shows a noticeable improvement compared to the prior and $y_{atm}$ in reducing the mean biases and root-mean-square-error (RMSE). As a result, the posterior reveals the spatial feature of the 'truth', namely a moister and colder cyclone center.

According to Eq. A5 and A6, the precision of this synergistic retrieval algorithm is given by the posterior uncertainty $S$, which is shown in Fig. A2. The posterior uncertainty is within 1 K in temperature and is around 0.5 ppmv in water vapor around
80 hPa. While the uncertainty in IWC is equivalent to the DARDAR-Cloud product, the simulation experiment conducted by Feng et al. (2021) showed that the retrieval is able to reduce the mean biases and RMSE in collocated IWC products caused by issues such as footprint mismatch.

Compared to other current satellite observational products in the UTLS, this retrieval has several advantages. First, the relatively small sampling footprint (15 km, the same as the size of the AIRS instantaneous FOV in the nadir) compared to limb-viewing sounders (>100 km) is beneficial for capturing small-scale variability directly impacted by deep convection. Second, the ability to retrieve temperature and water vapor above storms simultaneously. Third, the ability to retrieve atmospheric profiles near the cloud top. The simulation experiment conducted by Feng et al. (2021) evidenced that the synergistic method is capable of sounding the temperature profile near and slightly below the cloud top (within cloud optical depth of 1), while other products may not perform all-sky retrieval (AIRS L2, Susskind et al., 2003) or may be degraded by cloud presence (MLS v4.2 Schwartz et al., 2008; Livesey et al., 2017).

As depicted in the simulation experiment, the synergistic retrieval reveals the spatial variation in temperature and humidity. The retrieval is sensitive to the vertical variation of temperature (Fig. A3 (a,b)) but is not as sensitive to the vertical variation of water vapor (Fig. A3 (c,d)). The vertical resolution of water vapor retrieval is coarse because the mid-infrared channels are less sensitive to the dry stratosphere, leading to a strong smearing effect of the averaging kernel (Feng and Huang, 2018). This smearing effect is illustrated in Fig. A3. In this test, we increase the temperature at every 20 hPa interval by 5 K as shown in Fig. A3 (a), this corresponds to the term $x_t - x_0$ in Eq. A4. Similarly, the water vapor mixing ratio at every 20 hPa interval is increased by 50 % (considering the water vapor radiative effect is logarithmically scaled), as shown in Fig A3 (c). The responses from the averaging kernel are then calculated using Eq. A4, which are shown in Fig. A3 (b,d) for temperature and water vapor, respectively. Figure A3 (a,b) shows that the retrieved temperature responds well to perturbation at different vertical ranges. However, Fig. A3 (c,d) shows that the fine-scale water vapor perturbation would result in vertically broad, bottom-heavy water vapor anomalies in the retrieval. Nevertheless, the retrieval determines the changes in LIWV properly to detect (de)hydration. This is verified by the tests illustrated in Fig. A3 (c). Here, we prescribe random hydrations or dehydrations in randomly selected 20 hPa thick layers between 20 and 100 hPa, following the same pattern in Fig. A3 (c), for 1000 cases. We find that the LIWV changes produced by averaging kernel approach the truth better at higher altitudes, suggesting that the retrieval is more sensitive to perturbations at higher altitudes.

To evaluate the effects of cyclones on water vapor, samples above cyclones are compared to the climatology computed from the multi-year monthly mean of MLS data at the same grid as the retrieval samples. To remove the systematic bias caused by the higher vertical resolution of the MLS product, the MLS climatology is converted to the same vertical resolution using the averaging kernel of the synergistic retrieval, following Eq. A The mean of the converted climatology profiles at retrieved sample locations is shown in Fig. 9 (b) black line, while the mean of retrieved water vapor anomalies in comparison with this converted climatology is shown in Fig. 8 (d) and Fig. 9 (d). These anomalies measure the impacts of cyclones. We note that this bias correction procedure affects the vertical structure of anomaly in water vapor, but has a negligible impact on the anomaly in LIWV. Therefore, the conclusion on the hydration and dehydration impacts as shown in Fig. 10 are robust.

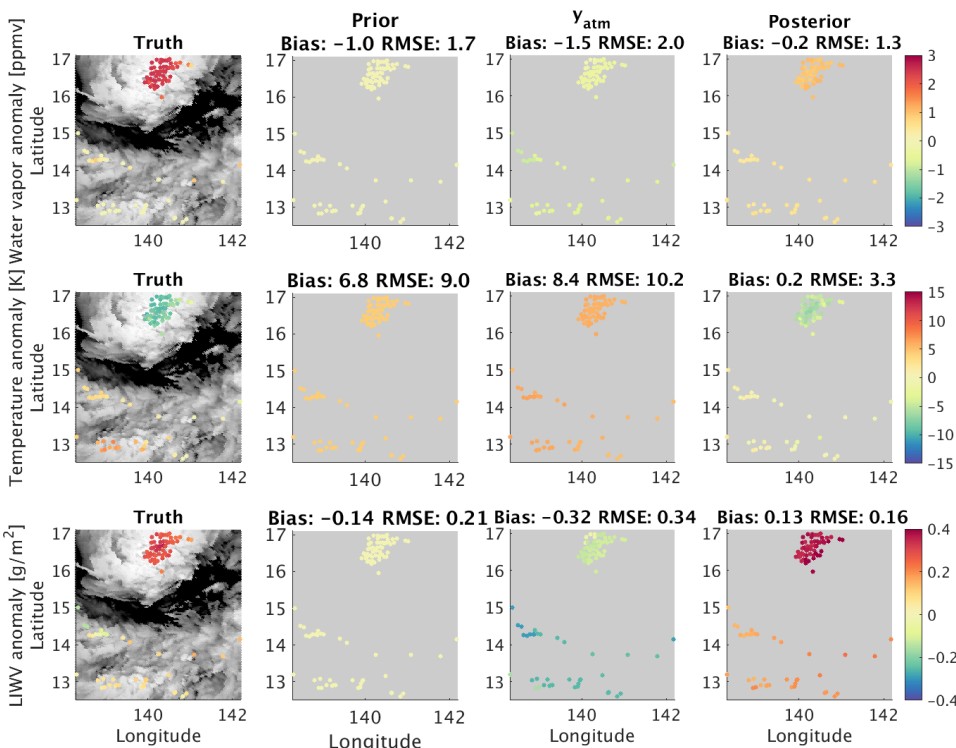

**Figure A1.** Horizontal distributions of the anomalies, defined as the deviation from the all-sky mean of the simulation field, in water vapor (in the units of ppmv, upper panels), temperature (in the units of K, middle panels) at 81 hPa, and layer integrated water vapor between 110 and 70 hPa (in the units of $g/m^2$, lower panels). The truth fields are shown in the first column, with its background grey-shaded for $BT_{1231}$. The other columns show the distribution in the prior, nearest ERA5 ($y_{atm}$), and the posterior of the retrieval.

## Appendix B: Binary classification of overshooting deep convective clouds

Similar to previous studies (Aumann and Ruzmaikin, 2013), we investigate BT of window channel at wavenumber $1231 \ \mathrm{cm}^{-1}$ ($BT_{1231}$) and water vapor absorption channel at wavenumber $1419 \ \mathrm{cm}^{-1}$ ($BT_{1419}$). In this section, we use collocated AIRS radiance observations and the DARDAR-Cloud product to evaluate the two metrics quantitatively and to identify the best threshold for determining the DCC-OTs.

   As shown in Fig. B1, the distributions of $BT_{1231}$ and $\Delta BT$ of overshooting DCCs resemble the Gaussian distribution in the

intervals of [185 210] K and [-2 12] K, respectively. To find the optimized threshold for DCC-OT classification, we calculate the accuracy and the f1 score of DCC-OT classification combining $BT_{1231} <= \varepsilon_{BT}$ and $\Delta BT >= \varepsilon_{\Delta BT}$. The accuracy ($\alpha$)

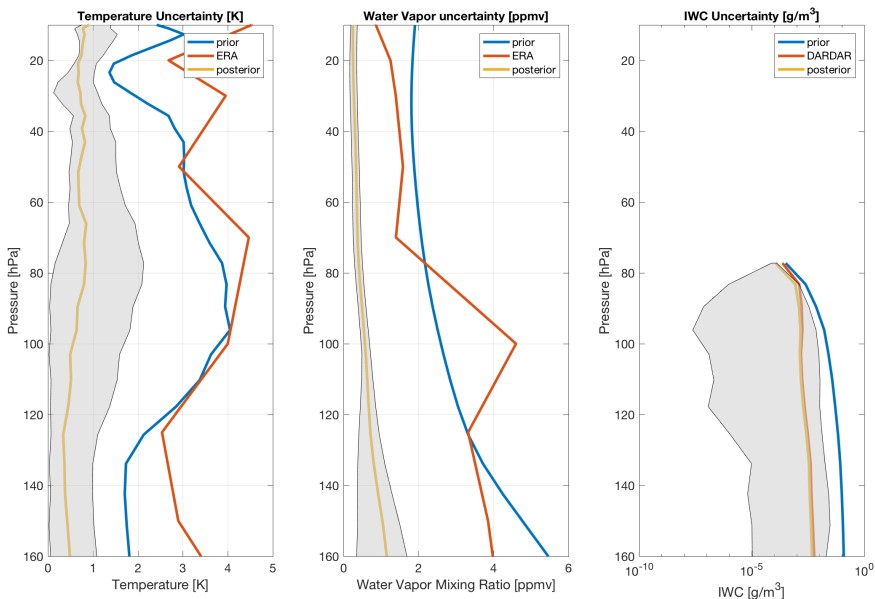

**Figure A2.** Uncertainties in temperature, water vapor mixing ratio, and IWC, estimated from the average of 2735 retrieved profiles with varied cloud top heights. Blue, red, and yellow curves show uncertainties of the prior ($S_a$), the ERA5 ($S_{model}$), and the posterior ($S_{post}$), respectively. The grey shaded area is the range of posterior uncertainties.

and the f1 score ($F1$) are defined as:

$$\alpha = \frac{TP+TN}{TP+FP+FN+TN};$$
$$P = \frac{TP}{TP+FP};$$
$$R = \frac{TP}{TP+FN};$$
$$F1 = 2 \times \frac{R \times P}{R+P};$$

(B1)

Where the TP, TN, FP, and FN are the number of true positive, true negative, false positive, and false negative, respectively. While the $P$ and $R$ represent the precision and recall (equivalent with true positive rate) of this classification, the f1 score considers both by a harmonic average of the two factors.

As indicated by Fig. B1 (c), the accuracy of the classification gets around 0.985 when $\varepsilon_{BT} <= 204$ K or $\varepsilon_{\Delta BT} >= 2$ K. However, this high accuracy is partly a result of a small sample size from DCC-OT compared to the total. The f1 score is
therefore used instead. The maximum $F1$ appears when $\varepsilon_{BT} = 203$ K and $\varepsilon_{\Delta BT} = 1$ K, although adding $\varepsilon_{\Delta BT}$ criterion only increase $F1$ by 0.0004 which is considered to be negligible here. Using $\varepsilon_{BT} = 203$ K leads to an FP rate at 0.008 and an FN rate at 0.323. Figure B1 (b) shows that the FP mainly comes from MIX and CI; their cold brightness temperature signal indicates sources from thick anvil cloud near the edge of the DCC system.

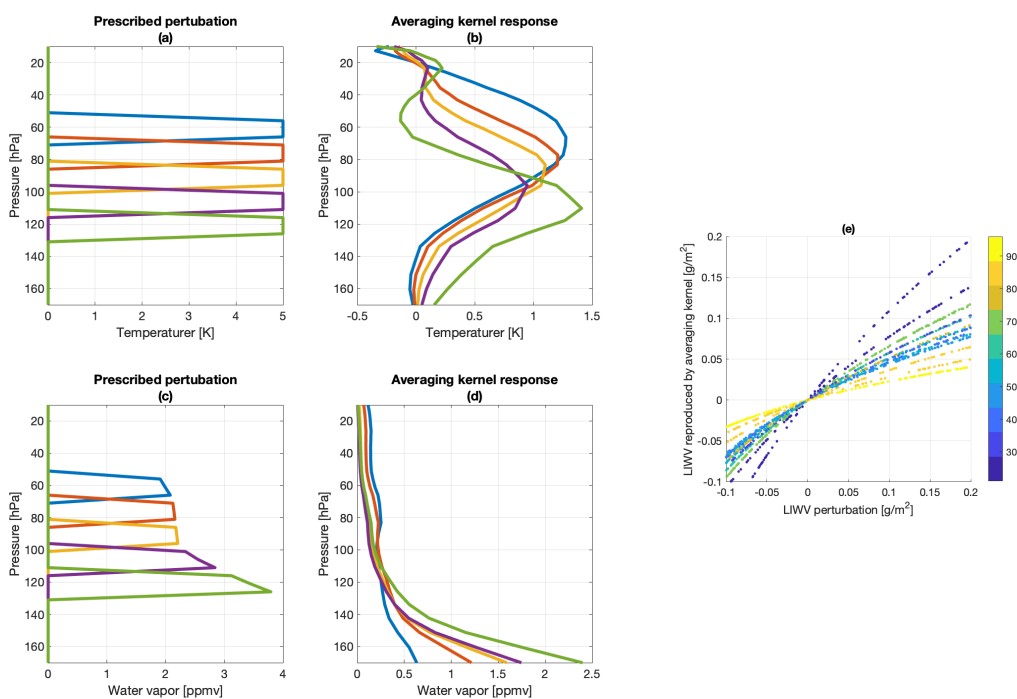

**Figure A3.** Averaging kernel responses to temperature and water vapor perturbation in thin layers. (a) Prescribed perturbation and (b) the response of averaging kernel in temperature [K]. (c) Prescribed perturbation and (d) the response of averaging kernel in water vapor [ppmv]. (e) LIWV perturbation [g/m²] between 100 and 20 hPa reproduced by averaging kernel, color-coded for vertical pressure intervals (hPa) where the perturbation is prescribed.

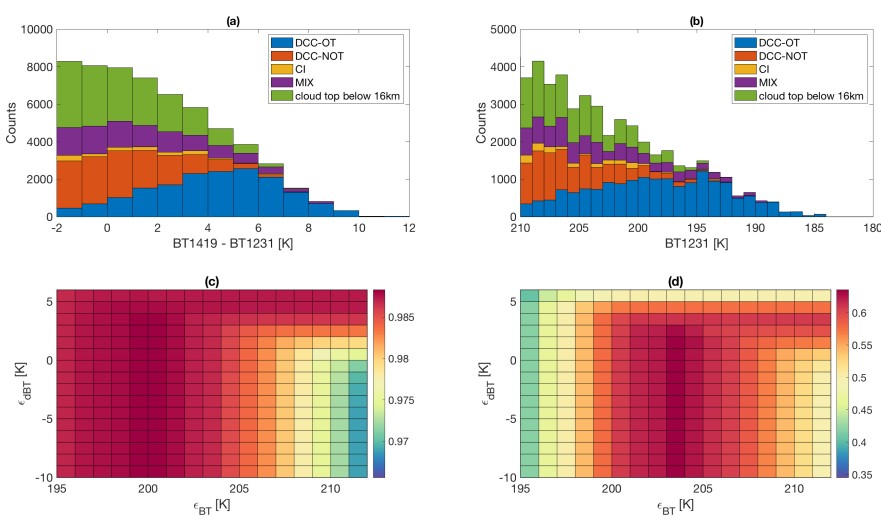

**Figure B1.** Distribution of (a) $BT_{1419}$-$BT_{1231}$ and (b) $BT_{1231}$ of AIRS FOVs for four TTL cloud categories and NTTLs (cloud top below 16 km). (c) The accuracy (Eq.1, $\alpha$) and (d) f1 score ($F1$) of the DCC-OT classification using $BT_{1231} <= \varepsilon_{BT}$ and $BT_{1419}$-$BT_{1231} >= \varepsilon_{\Delta BT}$ criterion.

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
