# Peer review of "Impacts of tropical cyclones on the thermodynamic conditions in the tropical tropopause layer observed by A-train satellites"

_Atmospheric Chemistry and Physics, 2021_

## Referee Comment (RC2)

Brief Initial Review of "Impacts of tropical cyclones on the thermodynamic conditions in the tropical tropopause layer observed by A-train satellites", by authors Feng and Wang

General: The authors have chosen a topic, modification of the thermodynamic structure of the tropopause by organized convective systems, that is both timely and interesting. This is an important topic because an accurate understanding of diabatic convective processes at the tropopause in the tropics and subtropics could impact the modeling of climate response to increased convection caused by rising surface temperatures. The analysis uses retrievals of temperature and water vapor profiles from AIRS, combined with a Radar-Lidar estimation of cloud ice water content to address an old and on-going debate about whether overshooting convective plumes can hydrate the lowermost stratosphere.  It is refreshing that the authors have provided an alternative to the very coarse-resolution MLS water vapor profiles to address this question, with the strong benefit of having a co-located retrieved temperature profile. The cyclone-centered coordinates are a welcome way to organize the observations.  I am recommending that this paper be accepted with some minor changes that are listed below.  In addition, vertical resolution around the tropopause for each data set/retrieval needs to be stated explicitly, some references ought to be consulted and updated to reflect the most current thinking and the development/uncertainty of satellite algorithms, for DARDAR-cloud and for the brightness temperatures.  Uncertainties and potential sampling biases need to be discussed. Are the retrievals representative?  How many AIRS-DARDAR combined profiles did not converge (I am only finding the number that did converge).  What do the authors think about diurnal changes?  Are these relevant to their results?  The choice of 16 km as a threshold for "overshoots" seems arbitrary and creates awkwardness for the interpretation because it includes the cold point.  Many tropical cyclones extend much higher than this at their cores, sometimes to 18 km. I'm not recommending that the authors redo the analysis, but acknowledgement that "overshoots" may instead be "cloud tops" ought to be included. Presentation of the new AIRS retrieval technique adds information and shows good promise, and it would be good to also see the authors present and understand limitations of this technique.

Specific suggestions:
The reference to Jensen 2007 should be updated to include more recent Ueyama et al. (2020) and Schoeberl et al. (2018) references in JGR.

The authors need to list version numbers for each of the data sources, including DARDAR. CloudSat does not observe small ice particles, and so cirrus cloud anvil edges with small effective particle size (< 40-50 microns) won't be included in the CloudSat data, but will be included in DARDAR.

17 km would likely be a better proxy for the tropopause at many locations in the tropics.  Check Tseng and Fu (2017) in JGR, for example and also for a discussion of the positive relationship between tropopause height and deep convection.  Cloud tops higher than 16 km are likely not

all really overshooting into the stratosphere, and in many places might include a local Ci maximum near the cold point tropopause.

Daytime and Nighttime differences in cloud top height for TTL CI in DARDAR could be substantial because of day/night differences in the Lidar observations. A large majority of TTL CI are only observed by the Lidar due to relatively small effective particle size. Do the authors mention whether they are analyzing daytime, nighttime or both?

To understand the difference between CI and MIX one needs to see Figure 4, so a recommendation is to reference this and to place it earlier in the paper. What is the anvil CI above DCC-NOT shown in this drawing? It appears to be part of the anvil, so would those profiles be CI or DCC-NOT?

Figures:
The Figure 1 caption is confusing. The sample density is for CALIPSO and CloudSat, both of which are used for DARDAR. If the sample density is measured at about 1x1 degree resolution (~ 100 km), how can it then be shown at higher resolutions? The text is more clear on this point, but a better caption would allow the figure to be understood better.

When discussing sampling it is appropriate to say CloudSat/CALIPSO because before 2015 they were both flying in formation in the A-Train, and both data sets are used in the DARDAR extinction retrieval and subsequent IWC estimation.

What is the vertical resolution of the combined AIRS-DARDAR temperature profile? What vertical resolution is the AIRS L2 and the MLS data converted to? It would be useful to know this in pressure, but also in equivalent geometric altitude.

---

## Author Comment (AC1)

**Response to Louis Rivoire, acp-2021-154**

near line 350: Figure 5(c) in Rivoire et al. (2020) illustrates the role of clear sky longwave radiation, which isn't directly relevant to a discussion about the effect of clouds on radiation. However, Figure 5(b) in Rivoire et al. (2020) could be referenced since it provides cloudy sky longwave radiative heating rates (most directly comparable to the first panel in Figure 11 in the present manuscript).

**Yes, it was a typo when we refer to Figure 5 in Rivoire et al. (2020). We now correct the reference to Fig. 5(b).**

Note that Figure 5(c) in Rivoire et al. (2020) makes the distinction between different kinds of cirrus-containing atmospheric columns, showing that longwave cirrus warming in the UTLS is strongest when cirrus are the only clouds in the column (which only happens in ~10% of the data set) and that the presence of other cloud types beneath UTLS cirrus clouds produces reduced cirrus warming or even cirrus cooling (which both occur in ~30% of the data set).

**Yes, and we separated different kinds of cirrus clouds in this manuscript as well. 'cirrus are the only clouds in the column' are 'CI' in our study. Depending on the underlying cloud type, other cirrus-containing columns can be 'DCC-NOT' or 'MIX'.**

**One difference between this study and Rivoire et al. (2020), however, is that I believe in Rivoire et al. (2020), an optical depth criterion is also used to identify thin cirrus. It is reasonable that after selecting thin cirrus-only columns, the longwave warming effect stands out as seen in Figure 5 (b) in Rivoire et al. (2020).**

I think that the paragraph starting at line 348 could be rephrased a bit to reflect these comments. For instance, the statement that no cirrus warming is found in Figure 11 could be rephrased since Figure 11 does not isolate cirrus effects, and also given that longwave warming does occur near and just above 100 hPa outside the 200 km radius in Figure 11, where cirrus clouds are very frequent (see e.g. Figure 3(a) in Rivoire et al., 2020).

**This paragraph has been rephrased. However, we would also like to clarify that:**

1) **Weak, positive HR indeed shows up in Figure 11 (a) but at the same altitude/distance range, the long HR is always smaller than the clear-sky HR, meaning that clouds have a cooling effect.**

2) **The statement at line 348 (original manuscript) does not intend to say that cirrus does not have a warming effect; instead, it simply stresses that the warming effect of thin cirrus is not visible when shown as a function of radial distances, because it is compensated by cooling effects of other cloud types.**

The statement that longwave radiative heating rates are mostly invariant with radius could also be nuanced since there is a vertical and radial dependence, and since Rivoire et al. (2020) arrived at similar results but also noted the strong dependence of cloud radiative effects on cloud type and cloud type combinations (similar to Figure 12 here), which show a radius dependency in tropical cyclones.

**I think the conclusion in cloud HR effects in this manuscript does not contradict Rivoire et al. (2020). There is indeed a very strong dependence of HR on cloud type, essentially determined by the vertical distribution of cloud optical thickness. Fig. 3 in Rivoire et al. (2020) shows that cloud types are strongly dependent on radial distances. However, such radial dependences become much weaker in Fig. 6 Rivoire et al. (2020), because the HR of different cloud combinations compensate at each radius bin. For cloud types with compensating HR effects (MIX), we tried to generate a plot similar to Figure 11 and find that the sign of HR effects is not clearly separated by radial distances.**

**One may estimate the cloud HR effects by constructing a careful classification of clouds using cloud optical depth profiles (i.e., thickness, position, and overlap of cloud layers). But please note that such cloud classification is only possible when active sensors like CloudSat/CALIPSO are available. Brightness temperature (BT) can be obtained from geostationary satellites that have much higher spatial and temporal coverage and BT of the window channel is sensitive to cloud thermal emission, which is impacted by optical thickness and the vertical cloud position. Therefore, we would like to point out that using BT is a direct and effective way to estimate the sign of HR at TTL.**

---

## Author Comment (AC2)

**Response to Review Comment 2, acp-2021-154**

This is a very thorough study of a relevant topic, with a number of interesting results. The analysis is careful and detailed, and the manuscript is well organized and well written. For these reasons I recommend publication with minor comments.

**Thank you for your positive comments on this manuscript. Following your suggestions, we rewrote the abstract and greatly expanded the Section 2 to briefly describe the synergistic retrieval and A-Train instruments used in this study.**

Here are those minor comments:

The use of CIWV to represent water vapor amount above a certain layer is non-standard and thus a bit confusing. 'Column integrated' implies total in the entire column; a web search shows many examples of this usage: https://www.google.com/search?client=firefox-b-1-d&q=column+integrated+water+vapor. One suggestion is to use "Layer Integrated Water Vapor", whose meaning should be immediately apparent.

**Done. Thank you for pointing it out.**

A few more details on the analysis are needed in the Abstract. Please explicitly mention the A-Train instruments used and what quantities are examined. Also, please expand briefly on the synergistic retrieval. What specific instruments does it use? A sentence or two should suffice. Also, how are cooling rates estimated. Again, a sentence or two will suffice.

**Following this suggestion, we rewrote the abstract and added necessary descriptions to instruments and methods.**

Finally, change 'convections' to 'convection'.

**Done.**

Line 118. Please state which two (or more?) AIRS L1B frequencies are used to estimate brightness temperature.

**Done.**

Include the OT, NOT, etc. definitions in figure 2 caption. Otherwise, terms for quantities in the figure cannot be fully understood without knowing where in the text the figure is discussed.

**We have re-order the figures and corresponding texts so that the schematic plot and the definition show up earlier.**

Line 339. Change 'cools' to 'cool'.

**Rephrased.**

Line 463. Change "under overcast cloud conditions" to "for overcast cloud conditions" or something similar. Please don't use "under" because it implies beneath the clouds.

**Done.**

---

## Author Comment (AC3)

**Response to Review Comment 1, acp-2021-154**

General: The authors have chosen a topic, modification of the thermodynamic structure of the tropopause by organized convective systems, that is both timely and interesting. This is an important topic because an accurate understanding of diabatic convective processes at the tropopause in the tropics and subtropics could impact the modeling of climate response to increased convection caused by rising surface temperatures. The analysis uses retrievals of temperature and water vapor profiles from AIRS, combined with a Radar-Lidar estimation of cloud ice water content to address an old and on-going debate about whether overshooting convective plumes can hydrate the lowermost stratosphere. It is refreshing that the authors have provided an alternative to the very coarse-resolution MLS water vapor profiles to address this question, with the strong benefit of having a co-located retrieved temperature profile. The cyclone-centered coordinates are a welcome way to organize the observations. I am recommending that this paper be accepted with some minor changes that are listed below.

**Thank you for your general assessments of this manuscript and constructive comments that help us improve. Following your suggestions, we reorganized the figures, improved the Section 2.1 by adding detailed introduction to each instrument and dataset, and included comparisons of MLS v5 and the synergistic retrieval in the Conclusion section.**

In addition, vertical resolution around the tropopause for each data set/retrieval needs to be stated explicitly,
**Done.**

some references ought to be consulted and updated to reflect the most current thinking and the development/uncertainty of satellite algorithms, for DARDAR-cloud and for the brightness temperatures.

**Done. Following this comment, Section 2.1 has been improved and expanded.**

Uncertainties and potential sampling biases need to be discussed.

**We account for uncertainties caused by sampling differences mainly by increasing the uncertainty range estimated by different products in the optimal estimation method. It is detailly discussed in the Appendix. Following this comment, we also add relevant information to Section 2.1.**

Are the retrievals representative?

**The retrievals are representative of atmospheric states above thick high clouds surrounding tropical cyclone centers and the composite is representative of each cloud category.**

**However, there might not be sufficient samplings in every radius bin. As a result, the composite constructed with retrieved values as a function of radial distances may not be representative of the geographical pattern associated with tropical cyclone events.**

How many AIRS-DARDAR combined profiles did not converge (I am only finding the number that did converge).

**Done.**
**3475 FOVs meet the criterion of 1) cold scenes 2) CloudSat footprints within 6.5 km from the center of FOV. 2735 converges. 740 FOVs do not converge. A typical situation for these rejected FOVs is that the radiance residual at the initial time step is too large (i.e., > 20K). It happens when cloud amount among a FOV is not uniform so that there is a large difference in cloud states between CloudSat (1.4 x 1.8 km) and AIRS footprint (13.5 x 13.5 km). It may also happen when the optical depth of the topmost cloud layer is less than 1 (in CI and MIX category). We assume that spectral optical properties with respect to cloud ice mass are uniform through vertical layers of an atmospheric column; this assumption fails when the topmost cloud layer does not effectively attenuate infrared radiation.**

What do the authors think about diurnal changes? Are these relevant to their results?

**We do notice statistically significant day-night differences. A-Train satellite overpass the same region two times a day, once in ascending nodes and the other in descending nodes. The attached figure shows nighttime minus daytime in (a) TTL cloud occurrence, DCC occurrence, and TTL cloud water, (b) BT1231 anomaly, and (c) ice water content, computed as the difference between descending nodes (nighttime) minus ascending nodes (daytime), based on the DARDAR-Cloud (a,c) and the AIRS L1B products (b). Sample densities are displayed in Fig.1 (b,d). Only statistically significant ($99\%$) differences are shown. Note there is a lack of nighttime observation operated by CloudSat after the year 2011, therefore, only overpasses between 2006-2011 are used for generating this figure. Due to a lack of sampling, day-night contrast in temperature and water vapor during this period retrieved by the joint AIRS-DARDAR method does not pass the significant test.**

**What we see here is that the nighttime overpasses are associated with higher TTL cloud occurrence frequency but lower IWC, especially in regions away from cyclone centers; higher DCC occurrence frequency and higher TTL cloud ice mass near cyclone centers. The BT1231 is warmer in the nighttime.**

**Although the day-night differences pass the significant test, we are not sure whether it is partly caused by differences in observations and retrieval process, for example, CALIPSO is calibrated using nighttime profiles and daytime calibration are interpolated from the nighttime, and 532-nm channel might be more sensitive to clouds during nighttime (Winker et al., 2009) so more nighttime TTL clouds could be a result of the higher sensitivity. We are not sure whether the contrast between ascending and descending nodes is truly representative of the diurnal cycle because the local solar time of overpasses is fixed at roughly 1:30 and 13:30. We find it curious that while there is a higher occurrence frequency of TTL clouds in the nighttime, the BT1231 is warmer (indicates lower cloud top altitude or less cloud cover) and IWC is lower away from cyclone centers. We do not find a rational way to relate the day-night difference with other results presented in this study, for example, the radiative heating with or without the shortwave heating. Therefore, we chose not to elaborate on the diurnal differences in this manuscript.**

[Figure]

The choice of 16 km as a threshold for "overshoots" seems arbitrary and creates awkwardness for the interpretation because it includes the cold point. Many tropical cyclones extend much higher than this at their cores, sometimes to 18 km. I'm not recommending that the authors redo the analysis, but acknowledgement that "overshoots" may instead be "cloud tops" ought to be included.

**We chose 16 km as a threshold because the averaged LZRH is located at 15.7 km and the 380 K potential temperature is located at 15.9 km, at the selected region (northern part of the west pacific). We focus on the potential temperature and LZRH, instead of the level neutral buoyancy, for identifying the threshold for 'overshoot', due to the focus on diabatic transports across isentropic surfaces in this study. We do not directly use potential temperature as a threshold because this variable is derived from ECMWF while one of the arguments of this study is that the reanalysis product may not be reliable in the TTL when deep convective storms occur.**

**You are right that using 'overshoot' can be misleading, as it refers to energetic convective events penetrate the level of neutral buoyancy, if not otherwise stated. In this manuscript, we use 'overshoot' to describe the continuous convective clouds that extend above 16 km (~ 380 K), which is the lower bound of the TTL as defined in this study. We also reword the definition of 'overshoot' to avoid confusion.**

Presentation of the new AIRS retrieval technique adds information and shows good promise, and it would be good to also see the authors present and understand limitations of this technique.

**Done. We also summarize and compare the limitation of the synergistic AIRS-based retrieval and MLS. The most noticeable drawbacks of the joint-AIRS-DARDAR retrieval method are 1) very limited samples as we only retrieve very thick high clouds currently, making further analysis of the convective impacts difficult; this can be overcome in the future, considering DeSouza-Machado et al. (2018) and Irion et al. (2018) have made some progresses in**

**incorporating passive sensors to retrieve single-FOVs in all-sky conditions. 2) strong smearing effect limited by spectroscopy of mid-infrared sensors that are not very sensitive to the dry stratosphere. 3) due to the limited precision (0.31 K for temeprature and 0.36 ppmv water vapor), we do not recommend to results from joint AIRS-DARDAR to analyze relative humidity.**

Specific suggestions:
The reference to Jensen 2007 should be updated to include more recent Ueyama et al. (2020) and Schoeberl et al. (2018) references in JGR.
**Done.**

The authors need to list version numbers for each of the data sources, including DARDAR. CloudSat does not observe small ice particles, and so cirrus cloud anvil edges with small effective particle size (< 40-50 microns) won't be included in the CloudSat data, but will be included in DARDAR.

**Done. Yes, radar is more sensitive to the largest particles within a cloud volume. To avoid this inconsistency, we use IWC from DARDAR and fluxes, heating rates, and cloud classifications from CloudSat 2B-*-LIDAR products (not the radar-only products), where CALIPSO observations are used as well. McErlich et al., 2020 compared cloud occurrences between 2B-CLDCLASS-LIDAR R05 and DARDAR v2.1.1 and found that DARDAR tends to have less cloud occurrence than 2B-CLDCLASS-LIDAR R05 at high altitudes. They found that DARDAR results agree better with ground-based observations from an AWARE campaign in Antarctica (although such ground-based campaign may underestimate high clouds). For the tropical high altitude in this study, we choose to use 2B-CLDCLASS-LIDAR R05 for cloud classification and DARDAR-Cloud v2.1.1 for cloud occurrence, as described in Section 3.1. We note that for tropical high clouds near tropical cyclone events, using 2B-CLDCLASS-LIDAR directly (without combing DARDAR data for adjustment of cloud occurrence) does not visually impact the result presented in the manuscript (Fig. 2, 3, 9, and 10.).**

17 km would likely be a better proxy for the tropopause at many locations in the tropics. Check Tseng and Fu (2017) in JGR, for example and also for a discussion of the positive relationship between tropopause height and deep convection. Cloud tops higher than 16 km are likely not all really overshooting into the stratosphere, and in many places might include a local Ci maximum near the cold point tropopause.

**We agree that 16 km may not be high enough for the lower boundary of the stratosphere. However, we do not intend to use it to identify overshoots into the stratosphere. Instead, we use 16 km conservatively to identify the upper-troposphere (lower part of TTL). As stated earlier in this response, the 16 km threshold is set considering the average altitude of LZRH and 380 K for regions within 1000km to cyclone centers over the northern part of the West Pacific, based on CloudSat-2B-FLXHR-LIDAR and CloudSat-ECMWF-AUX. The 380 K isentropic surface may be located at a higher altitude when perturbations in thermodynamic conditions are large, but we do not have a comprehensive temperature dataset with high quality to identify it.**

**Following this comment, we expand Section 3.1 to clarify the source of CIs. It can be formed by 1) transport of cloud ice via convective outflow or propagating wave activities, and by 2) local cooling that condensates supersaturated vapor that is found in Tseng and Fu (2017) in JGR, where the cooling can be a result of dynamical or radiative processes. This study does**

**not intend to attribute the source of CI above 16km to convective overshoot. Instead, we break down the TTL cloud ice from DCC-OTs and CIs separately and try to highlight that during tropical cyclone events the total mass of cloud ice above 16 km in CI type is negligible compared to those directly in the top of deep convective clouds (which we phrase as 'overshoot'), despite the frequent occurrences of CIs.**

Daytime and Nighttime differences in cloud top height for TTL CI in DARDAR could be substantial because of day/night differences in the Lidar observations. A large majority of TTL CI are only observed by the Lidar due to relatively small effective particle size. Do the authors mention whether they are analyzing daytime, nighttime or both?

**See previous responses. We include both. To clarify, we only use data when both radar and lidar observations are available so that the construct between ascend-descend is not a result of the lack of radar observations. Despite the diurnal differences mentioned earlier, we do not notice eyeballing differences in the composite by including/excluding data between 2011 to 2016, where only daytime is available.**

To understand the difference between CI and MIX one needs to see Figure 4, so a recommendation is to reference this and to place it earlier in the paper.
**Done.**

What is the anvil CI above DCC-NOT shown in this drawing? It appears to be part of the anvil, so would those profiles be CI or DCC-NOT?
**These profiles will be DCC-NOT because the 'CI' in our classification refers to high cloud-only above a clear troposphere. Any cloud column containing DCC and other cloud types is classified as DCC-NOT.**

Figures:
The Figure 1 caption is confusing. The sample density is for CALIPSO and CloudSat, both of which are used for DARDAR. If the sample density is measured at about 1x1 degree resolution (~ 100 km), how can it then be shown at higher resolutions? The text is more clear on this point, but a better caption would allow the figure to be understood better.

**Thanks for pointing it out. Take AIRS as an example, we count the number of samples at each 20 x 20 km first and then convert it to number per 100km x 100km so that the number density at different resolutions can be comparable (i.e., number density in Figure 1 (b-d) is in the same unit). We add the definition of the sample density and how it is calculated in the text.**

When discussing sampling it is appropriate to say CloudSat/CALIPSO because before 2015 they were both flying information in the A-Train, and both data sets are used in the DARDAR extinction retrieval and subsequent IWC estimation.
**Done.**

What is the vertical resolution of the combined AIRS-DARDAR temperature profile? What vertical resolution is the AIRS L2 and the MLS data converted to? It would be useful to know this in pressure, but also in equivalent geometric altitude.

**We added this information into the manuscript. The vertical resolution is 3.2 km for temperature and 5.8 km for water vapor. Please note that the retrieved values are provided roughly every 0.4 km around the tropopause; the vertical resolution is the FWHM of the vertical averaging kernel that the joint AIRS-DARDAR method can resolve.**

---

## Referee Report (RR1)

Comments/Corrections to acp-2021-154-ATC2:

Page 4, lines 95-100:
This isn't stated correctly. The Lidar does not measure particle concentrations, and can observe below the highest clouds if they are thin. The Radar doesn't measure clouds with small particle sizes in a profile (so it doesn't always provide the full profile).

Suggest you state this instead:

The lidar signals are sensitive to cloud particles at all sizes but are quickly attenuated in thick clouds. The radar is not sensitive to clouds with effective particle sizes smaller than 40-50 microns, but can observe thick clouds and precipitation in storm cores.

Page 4, around line 118

It's good to mention that you are using DARDAR v2.1.; this version of DARDAR uses CALIPSO Version 3 and CloudSat R04, both of which have been replaced by newer, more accurate versions of the data. I think that it is good to mention which versions of the Radar and Lidar are being used in this study (and it is easy to add this information here).

Page 6, end of page (last sentence)

This is a good place to mention that any diurnal variability that may be occurring is not being addressed in this study. It's OK not to address it, but it's important to acknowledge it.

Abstract/Introduction/Section 3.1

This information is important and needs to be stated BEFORE the term overshoot is ever used. It needs to be in the abstract as well.

Your reply to my review states, "**We chose 16 km as a threshold because the averaged LZRH is located at 15.7 km and the 380 K potential temperature is located at 15.9 km, at the selected region (northern part of the west pacific)."**

This is good and important information, where is this stated explicitly and how are we to interpret "overshoot" before we are given this info?

Conclusion: Did you add this to the paper?

**"However, there might not be sufficient samplings in every radius bin. As a result, the composite constructed with retrieved values as a function of radial distances may not be representative of the geographical pattern associated with tropical cyclone events."**

That is important to know when interpreting the results.

Here is one more important thing to state in the paper:

**"3475 FOVs meet the criterion of 1) cold scenes 2) CloudSat footprints within 6.5 km from the center of FOV. 2735 converges. 740 FOVs do not converge. A typical situation for these rejected FOVs is that the radiance residual at the initial time step is too large (i.e., > 20K). It happens when cloud amount among a FOV is not uniform so that there is a large difference in cloud states between CloudSat (1.4 x 1.8 km) and AIRS footprint (13.5 x 13.5 km). It may also happen when the optical depth of the topmost cloud layer is less than 1 (in CI and MIX category). We assume that spectral optical properties with respect to cloud ice mass are uniform through vertical layers of an atmospheric column; this assumption fails when the topmost cloud layer does not effectively attenuate infrared radiation."**

It is critical to acknowledge that 20% of your retrievals won't close, and that this is selectively when the topmost cloud layer is optically thin.

---

## Author Response (AR3)

**Point to point responses, ACP-2021-154**

Page 4, lines 95-100:

This isn't stated correctly. The Lidar does not measure particle concentrations, and can observe below the highest clouds if they are thin. The Radar doesn't measure clouds with small particle sizes in a profile (so it doesn't always provide the full profile).

Suggest you state this instead:

The lidar signals are sensitive to cloud particles at all sizes but are quickly attenuated in thick clouds. The radar is not sensitive to clouds with effective particle sizes smaller than 40-50 microns, but can observe thick clouds and precipitation in storm cores.

**Thank you. This statement is better and more precise; we have added it to the manuscript.**

Page 4, around line 118

It's good to mention that you are using DARDAR v2.1.; this version of DARDAR uses CALIPSO Version 3 and CloudSat R04, both of which have been replaced by newer, more accurate versions of the data. I think that it is good to mention which versions of the Radar and Lidar are being used in this study (and it is easy to add this information here).

**Done. Thank you for pointing out that DARDAR v2.1 use the updated version of CALIPSO.**

Page 6, end of page (last sentence)

This is a good place to mention that any diurnal variability that may be occurring is not being addressed in this study. It's OK not to address it, but it's important to acknowledge it.

**Done.**

Abstract/Introduction/Section 3.1

This information is important and needs to be stated BEFORE the term overshoot is ever used. It needs to be in the abstract as well.

Your reply to my review states, "We chose 16 km as a threshold because the averaged LZRH is located at 15.7 km and the 380 K potential temperature is located at 15.9 km, at the selected region (northern part of the west pacific)."

This is good and important information, where is this stated explicitly and how are we to interpret "overshoot" before we are given this info?

Conclusion: Did you add this to the paper?

**This information appears at the beginning of Section 3.1, at L170 and L179 , before the definition of 'DCC-OT' to clarify how we select 'overshooting' clouds in this study. It might not be necessary to move it to the first appearance of 'overshoot(ing)' in the Introduction because this wording is more generalized in the introduction, and we later specified it in Section 3.**

"However, there might not be sufficient samplings in every radius bin. As a result, the composite constructed with retrieved values as a function of radial distances may not be representative of the geographical pattern associated with tropical cyclone events."

**Yes, this information was updated to the manuscript. Please check L284.**

That is important to know when interpreting the results. Here is one more important thing to state in the paper:

"3475 FOVs meet the criterion of 1) cold scenes 2) CloudSat footprints within 6.5 km from the center of FOV. 2735 converges. 740 FOVs do not converge. A typical situation for these rejected FOVs is that the radiance residual at the initial time step is too large (i.e., > 20K). It happens when cloud amount among a FOV is not uniform so that there is a large difference in cloud states between CloudSat (1.4 x 1.8 km) and AIRS footprint (13.5 x 13.5 km). It may also happen when the optical depth of the topmost cloud layer is less than 1 (in CI and MIX category). We assume that spectral optical properties with respect to cloud ice mass are uniform through vertical layers of an atmospheric column; this assumption fails when the topmost cloud layer does not effectively attenuate infrared radiation."

It is critical to acknowledge that 20% of your retrievals won't close, and that this is selectively when the topmost cloud layer is optically thin.

**Yes, thanks for pointing it out. It was discussed in L272. Sorry that we did not include the exact line numbers where our revisions were made in the previous reply.**